# The retrieval of previously learned motor memories is facilitated by the reinstatement of default mode network manifold structures

Ali Rezaei[1], Corson N. Areshenkoff[1,2], Daniel J. Gale[1], Emily R. Oby[1,3], Jonathan Smallwood[1,2], J. Randall Flanagan[1,2], Jeffrey D. Wammes[1,2], Jason P. Gallivan[1,2,3]*

**1** Centre for Neuroscience Studies, Queen's University, Kingston, Ontario, Canada, **2** Department of Psychology, Queen's University, Kingston, Ontario, Canada, **3** Department of Biomedical and Molecular Sciences, Queen's University, Kingston, Ontario, Canada

* gallivan@queensu.ca

## Abstract

Motor learning induces alterations in neural activity that can persist long after the effects of such learning have faded. These persistent neural alterations are thought to manifest behaviorally as "savings," or faster relearning, via access to a latent motor memory. How the human brain forms and retrieves these latent memories, and the specific neural systems involved, remains unresolved. Here, using human functional MRI and a two-day sensorimotor adaptation paradigm, we show that savings are associated with the reinstatement of a large-scale cortical manifold structure formed during initial learning. Notably, this neural reinstatement effect was not observed across sensorimotor systems but was localized to regions of the default mode network (DMN). Moreover, the specific dynamics of DMN activity were linked to inter-subject differences in patterns of learning and relearning across days. These results suggest that motor savings arises from the re-expression of DMN activity patterns associated with initial learning, establishing a key role for this network in motor memory formation and retrieval. This finding, paralleling reinstatement principles from other memory domains (episodic memory, fear conditioning) and anticipated by recent computational models of motor learning, suggests a common mechanism for the flexible recall and reuse of stored memories across diverse behavioral contexts.

## Introduction

The capacity to flexibly adapt our motor actions to changing environmental demands and retain those adaptations over time is fundamental for skilled behavior [1,2]. A classic hallmark of these retained adaptations is the phenomenon of "savings"—the observation that relearning a motor skill is faster upon subsequent practice than compared to the initial learning episode [3–6]. This "savings" effect, echoing Ebbinghaus'

**Data availability statement:** The underlying numerical data for figures are provided in S1 Data and archived on Zenodo (https://doi.org/10.5281/zenodo.18613054). Raw behavioral and imaging data (including T1w and functional scans) are openly available on OpenNeuro (https://doi.org/10.18112/openneuro.ds005598.v1.0.3). The analysis code is archived on Zenodo (DOI: https://doi.org/10.5281/zenodo.18612771).

**Funding:** This work was supported by operating grants from the Canadian Institutes of Health Research (PJT175012; https://cihr-irsc.gc.ca/) and the Natural Sciences and Engineering Research Council (RGPIN-2017-04684; https://www.nserc-crsng.gc.ca/), awarded to J.P.G. The funders had no role in study design, data collection and analysis, decision to publish, or preparation of the manuscript.

**Competing interests:** The authors have declared that no competing interests exist.

**Abbreviations:** CSF, cerebrospinal fluid; CW, clockwise; DMN, default mode network; FC, functional connectivity; FD, framewise displacement; fMRI, functional magnetic resonance imaging; fPCA, functional principal component analysis; GM, gray matter; mPFC, medial prefrontal cortex; PC, principal component; PCA, principal component analysis; PCC, posterior cingulate cortex; RR, Recall Ratio; UMAP, uniform manifold approximation and approximation; VMR, visuomotor rotation; WM, white matter.

early work on declarative memory [7], implies the existence of a "memory trace" or residual representation of the initial learning experience, which endures even when its effects are no longer evident in behavior [8–13]. Identifying the nature and physical substrate of this lingering memory trace—termed the "engram" by Semon more than a century ago [14]—remains a central challenge in neuroscience [15–19]. While recent studies in rodents have identified neuronal ensembles (engrams) critical for specific fear or spatial memories [15–17,20,21], how the human brain forms, retains and retrieves the complex sensorimotor memories that facilitate savings remains poorly understood.

Several theories have emerged within the motor learning field to explain the mechanism underlying savings. One perspective emphasizes the recall of a latent representation of the adapted motor state acquired during initial learning [3,22]. Recent computational models have formalized this as the re-expression of a specific, context-dependent motor memory upon re-exposure [23]. An alternative, though related, view emphasizes the rapid retrieval and re-implementation of explicit cognitive strategies developed during initial learning to facilitate adaptation [9,24,25]. Yet another proposal suggests that savings originates from a memory of past errors, leading to increased error sensitivity and faster error correction when those same errors are re-encountered [6]. Importantly, while these theories differ on the *exact* content of the retrieved memory—be it a motor state, cognitive strategy, or error sensitivity—they all converge on the core idea that savings involves the re-activation of information that persists from the initial learning period. This general view is consistent with evidence of persistent neural changes in rodent and nonhuman primate motor cortex following motor learning [26–28] and aligns with the broader cognitive neuroscience perspective that memory retrieval—at least declarative memory retrieval—involves the re-activation of distributed activity patterns associated with initial encoding [29–42]. This raises key questions about the neural implementation of motor memory retrieval: First, is motor savings actually associated with the reinstatement of neural activity patterns observed during the initial motor learning episode? Second, if so, is this process confined to sensorimotor areas [26–28] or does it involve higher-order networks implicated in processes such as memory retrieval and strategic control?

Although motor learning necessarily entails the modification of sensorimotor circuits [26–28], the substantial contribution of explicit strategies [9,43] and memory retrieval [6,22,23] suggests that brain systems beyond primary motor areas are involved [44]. Among these, the default mode network (DMN) emerges as a particularly compelling possibility. While historically linked to internally focused states [45–53], the DMN is now better understood as a system that supports memory-guided cognition, enabling the flexible application of task strategies [54–56] and the guidance of actions based on prior experience [57–60]. This functional role is not limited to cognitive tasks; indeed, emerging evidence implicates the DMN in motor behavior as well. For instance, both initial sensorimotor adaptation and its transfer to new contexts are underpinned by large-scale functional connectivity (FC) changes within the DMN [61–63]. However, while these findings link the DMN to the generalization of a learned skill, it remains an open question whether the DMN also

supports its retention and retrieval over time—the core process underlying savings. The DMN's neuroanatomy makes it well-suited for such a role. Its position at the apex of the cortical hierarchy allows it to integrate information across distributed brain systems [64], while its unique microstructure provides a flexible interface between current sensory inputs and retrieved internal states [65]. This architecture, seemingly optimized for reinstating prior experience, led us to hypothesize that faster relearning (savings) would be associated with the reinstatement of DMN activity patterns from the initial learning episode.

To explore the role that the DMN plays in motor savings, we used functional MRI (fMRI) to record whole-brain activity as human participants adapted their reaching movements during a classic visuomotor rotation (VMR) task across two consecutive days. This allowed us to measure whole-brain activity patterns during initial learning on Day 1 and subsequent faster relearning (savings) on Day 2. Recognizing that motor learning involves dynamic coordination across large-scale neural systems, potentially involving both sensorimotor and higher-order cognitive networks [2,44,66–69], we employed whole-brain manifold learning [64,70,71]. This approach characterizes changes in the macroscale organization of brain-wide FC by embedding the unique connectivity profile of each brain region into a common low-dimensional space, representing the brain's overarching functional geometry [64,72]. By tracking how brain regions change their position within this manifold space across task phases [61,73–76], we could identify large-scale network reconfigurations that are most strongly associated with initial learning and subsequent savings. In principle, regions important for motor savings should be important (1) when participants initially learn how to adapt their behavior during the VMR task, and (2) when they benefit from this learning in the subsequent session.

We show that the faster relearning characteristic of savings is marked by the reinstatement of a large-scale cortical manifold structure established during initial adaptation. Notably, this neural reinstatement was mainly observed in DMN regions, rather than being widely diffused across sensorimotor systems. Furthermore, we found that individual differences in subjects' patterns of learning and relearning across days correlated with the changes in the dynamics of DMN activity during the initial learning (and relearning) phases on both days. Together, these results suggest that, at the neural systems level, the retrieval of motor memories is supported by the reinstatement of the same DMN network activity patterns that underpin initial learning, providing a systems-level account of how the brain leverages past experience to advantage future behavior.

## Results

To investigate the neural processes underlying human motor adaptation and savings, we recorded whole-brain fMRI activity from participants ($N = 32$) performing a VMR task [77] across two testing sessions separated by 24 hours. In this center-out reaching task, participants used their dominant right hand to apply force to an MRI-compatible force sensor, controlling a cursor aimed at one of eight peripheral targets (Fig 1A). The first testing block began with 120 Baseline trials where cursor movement directly matched the direction of the applied force. This was followed by a 320-trial Learning block where a 45° clockwise (CW) rotation was imposed between the applied force and the cursor's trajectory, requiring participants to learn a compensatory counter-rotation. Finally, we used a 120-trial Washout block with no rotation to assess de-adaptation. Participants returned exactly 24 hours later (Day 2) and repeated the identical sequence of Baseline, Learning, and Washout blocks. This two-day design enabled us to directly compare adaptation versus de-adaptation, as well as initial learning (Day 1) versus subsequent relearning (Day 2), and thus examine the neural correlates of motor memory formation, persistence and retrieval.

### Behavioral signatures of motor learning and savings

Consistent with previous studies [2,25,77,78], participants successfully adapted to the 45° CW rotation on Day 1, gradually reducing their angular errors to near-Baseline levels, and exhibited the typical washout aftereffects (Fig 1B). Crucially, upon re-exposure to the rotation on Day 2, participants demonstrated significant behavioral savings at the group-level,

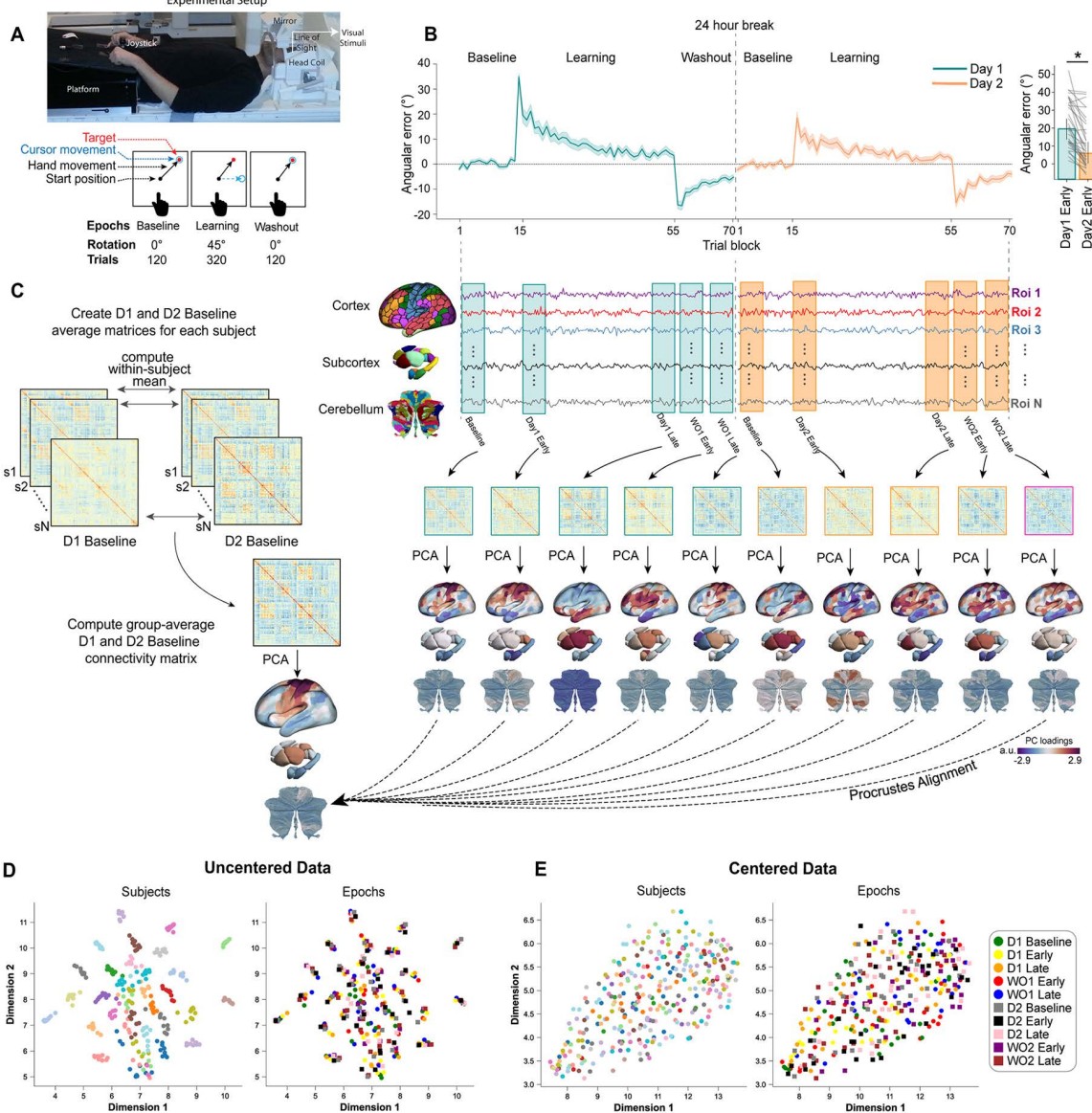

**Fig 1. Experimental design, behavioral results, and fMRI analysis overview. (A)** Participants performed center-out target-directed movements using an MRI-compatible force sensor (top schematic). The task structure (bottom) involved Baseline, Learning (45° CW rotation), and Washout blocks performed over two consecutive days (Day 1 and Day 2), allowing comparison between initial learning and relearning 24 hours later. **(B)** Group-averaged angular error (*N* = 32 participants) across 8-trial bins for Day 1 (cyan trace) and Day 2 (orange trace). Shading indicates ±1 SEM (standard error of the mean). Participants successfully adapted during Day 1 Learning and showed significant savings (faster relearning) on Day 2. Inset: A direct comparison of initial angular errors (first two 8-trial bins;16 trials) shows they were significantly lower on Day 2 than on Day 1 (*$p < 0.01$), confirming behavioral savings. Thin gray lines represent individual subject data. **(C)** Schematic of the neural analysis pipeline. Functional connectivity (FC) matrices were calculated for five specific task epochs on both days (colored boxes, right panel) using BOLD timeseries extracted from cortical, subcortical, and cerebellar regions (see Methods). Manifold learning techniques were applied to these FC matrices after a centering procedure (see Methods, see also panels **D** and **E**). The left panel illustrates the creation of a common template Baseline manifold, generated by averaging the centered FC matrices from Day 1 and Day 2 Baseline epochs across subjects. All individual epoch manifolds were then aligned to this common template using a Procrustes transformation. This created a unified space to assess learning- and savings-related changes relative to the baseline functional architecture. **(D and E)** Visualization of the effect of our centering procedure on FC matrix similarity using UMAP. Each point represents an FC matrix from one participant and epoch. **(D)** Before centering, matrices cluster primarily by subject (left panel, colored by subject), obscuring task-related structure (right panel, colored by epoch; see legend at bottom right). **(E)** After applying Riemannian centering, the subject-level clustering is eliminated (left panel), allowing the underlying task-related structure to become visible (right panel).

characterized by markedly faster error reduction compared to initial learning (Fig 1B, inset). This confirms the retention and successful retrieval of the motor memory formed during Day 1 [10,77].

## Neural analysis approach: Manifold representation of functional connectivity

To probe the large-scale neural changes accompanying motor memory formation, retention, and savings, we examined task-related FC across key task epochs on both days. Specifically, we defined five distinct, equal-length 48-trial (96 imaging volume) epochs for each day: Baseline, Early Learning, Late Learning, Early Washout, and Late Washout (yielding 10 epochs in total). For each participant and epoch, we extracted region-wise BOLD time series from a whole-brain parcellation comprising cortical [79], subcortical [80], and cerebellar regions [81] and then estimated whole-brain covariance (FC) matrices [for a similar approach, see 61,63,74,76].

Recognizing that stable, subject-specific factors in FC can obscure task-related variance [82,83], we first centered participants' covariance matrices using a validated Riemannian geometric approach [61,63,74–76,84,85]. This procedure isolates epoch-specific variations in FC by translating all individual FC matrices to a common group-average mean (see Materials and methods). The significant impact of this centering step is illustrated in Fig 1D and 1E using uniform manifold approximation (UMAP; [86]). Before centering (Fig 1D), FC matrices cluster primarily by subject identity, completely masking underlying task structure. After applying Riemannian centering (Fig 1E), this subject-level clustering is eliminated, allowing us to better explore any task-epoch related organization.

We then applied well-established manifold learning techniques to these centered matrices [71,87,88]. For this analysis, each centered covariance matrix was transformed into an affinity matrix, where values reflect the similarity between different regions' overall connectivity profiles. Principal component analysis (PCA) was then applied to these affinity matrices to identify the dominant low-dimensional axes of connectivity organization, which define the manifold or "functional geometry" of the brain for that specific epoch. To enable direct comparison across all epochs, days, and participants, each individual manifold was then aligned to a common template Baseline manifold (derived from the average of Day 1 and Day 2 centered baseline FC data) using Procrustes alignment. This unified space (Fig 1C, left) allowed us to identify systematic deviations from the baseline functional architecture during subsequent learning on Day 1 and relearning on Day 2.

## Manifold structure during baseline trials

The template Baseline manifold, derived from average FC data during baseline trials on both days, reveals the brain's large-scale functional organization during task performance. The first three principal components (PCs) captured a substantial portion (~46%) of the variance in baseline FC patterns (Fig 2A and 2B) and were retained for further analysis (however, note that inclusion of PCs 4–10 yielded qualitatively similar results; S1 Fig). PC1 primarily distinguished sensorimotor cortical regions and the putamen from the cerebellum and other cortical areas (Fig 2A). PC2 established a gradient between visual/cerebellar areas and higher-order association cortices, notably the DMN and frontoparietal control networks. PC3 largely differentiated visual and parietal cortical regions from subcortical and cerebellar structures. This task-based architecture is highly consistent with that observed in our prior work [61,63,74]

To quantify each brain region's embedding within this baseline architecture and track learning-related changes, we calculated its "manifold eccentricity" [61,63,73,74,76]. This metric is the Euclidean distance of a brain region from the manifold's center (origin) in the three-dimensional PC space, which captures the uniqueness of its FC profile (Fig 2D). Functionally, a region's eccentricity reflects its level of integration versus segregation: regions closer to the center (lower eccentricity) have more diverse, brain-wide connectivity profiles, whereas regions further from the center (higher eccentricity) are more functionally specialized and segregated within their own network (Fig 2E; [73,90,91]). This interpretation is supported by strong linear correlations between eccentricity and different graph theoretical metrics of segregation and integration (S2 Fig). Changes in eccentricity thus reflect relative shifts in regional integration and segregation during learning and savings.

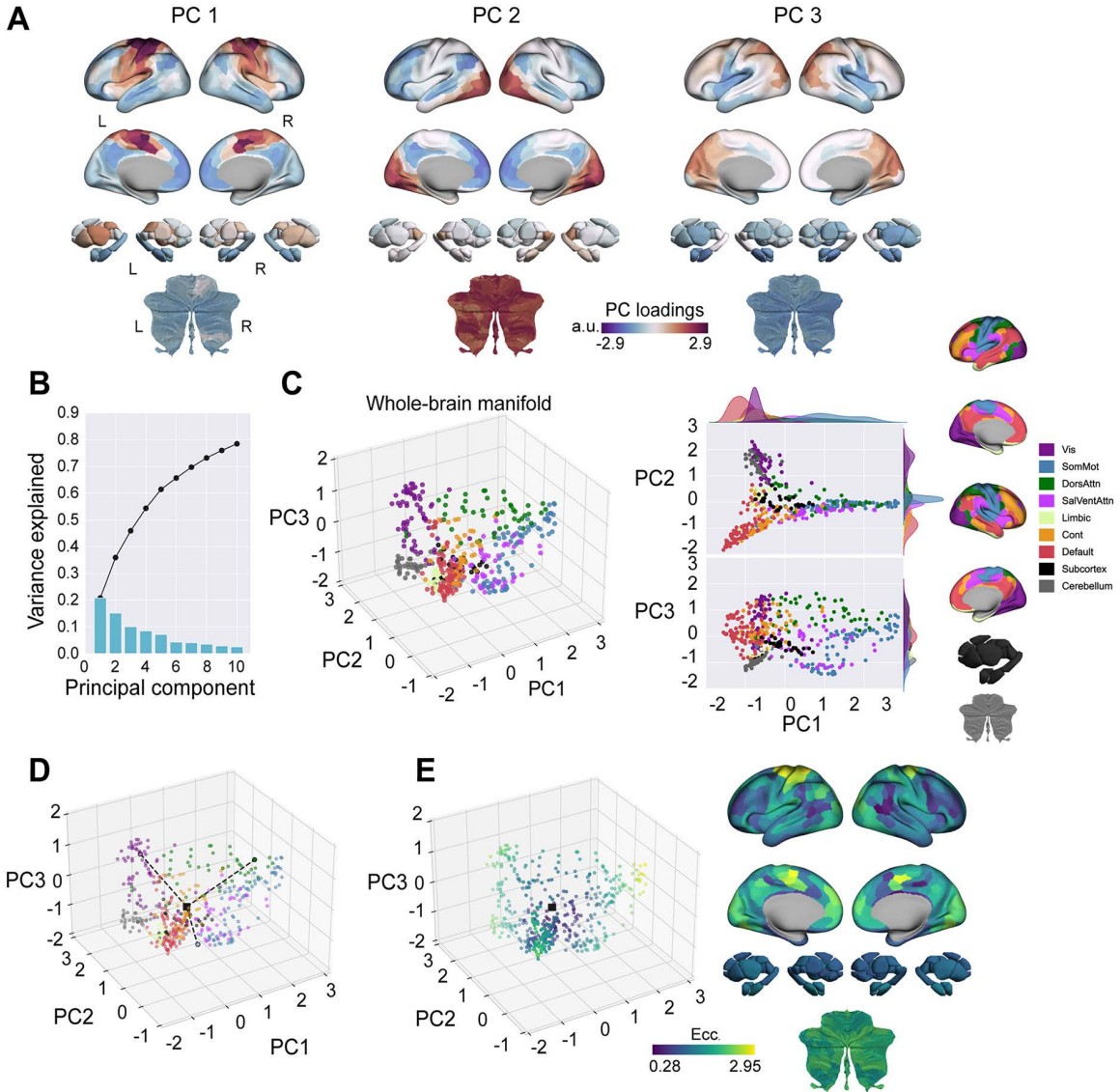

**Fig 2. Structure and eccentricity of the template Baseline manifold. (A)** Spatial distribution of loadings (component weights) for the top three principal components (PCs) derived from the template Baseline manifold, displayed across cortical, subcortical, and cerebellar surfaces. **(B)** Percentage of variance accounted for by the initial 10 PCs. The bars represent the contribution of each individual PC, while the black trace indicates the cumulative variance explained. **(C)** Visualization of the template Baseline manifold structure in low-dimensional (3 PC) space. Each point corresponds to a brain region, colored according to its assignment within the Yeo et al. 7-network parcellation [79,89]. **(D)** Schematic illustrating the calculation of manifold eccentricity. Eccentricity for a given region is computed as its Euclidean distance (dashed line) from the manifold's centroid (black square) in 3 PC space. Eccentricities for three representative regions are indicated with the dashed lines. **(E)** Baseline manifold eccentricity (ecc.) values are shown for all brain regions, color-coded and projected onto both low-dimensional manifold space (left) and the corresponding brain surfaces (right). On the manifold plot, the centroid is marked by a black square. L = left; R = right.

## Learning-dependent reconfigurations of manifold structure are consistent across days

To identify brain regions whose manifold embedding changed across the task, we performed a region-wise 2 (Day) × 5 (Task Epoch: Baseline, Early Learning, Late Learning, Early Washout, Late Washout) repeated measures ANOVA (rmANOVA) on manifold eccentricity. After FDR correction ($q < 0.05$), we found no significant main effect of Day or a Day ×

Task Epoch interaction in any brain region. Despite the potential for subtle session-to-session neural drift, this indicates that large-scale network reconfigurations associated with task performance were highly similar across Day 1 and Day 2. Consistent with this, we did observe a significant main effect of Task Epoch across several brain regions (98 cortical regions in total, see Fig 3A). These regions were distributed across multiple large-scale networks (Fig 3B), with the most prominent clusters occurring within the DMN, including the medial prefrontal cortex (mPFC) and posterior cingulate cortex (PCC), as well as areas in visual cortex and sensorimotor cortex. Importantly, control analyses showed that these

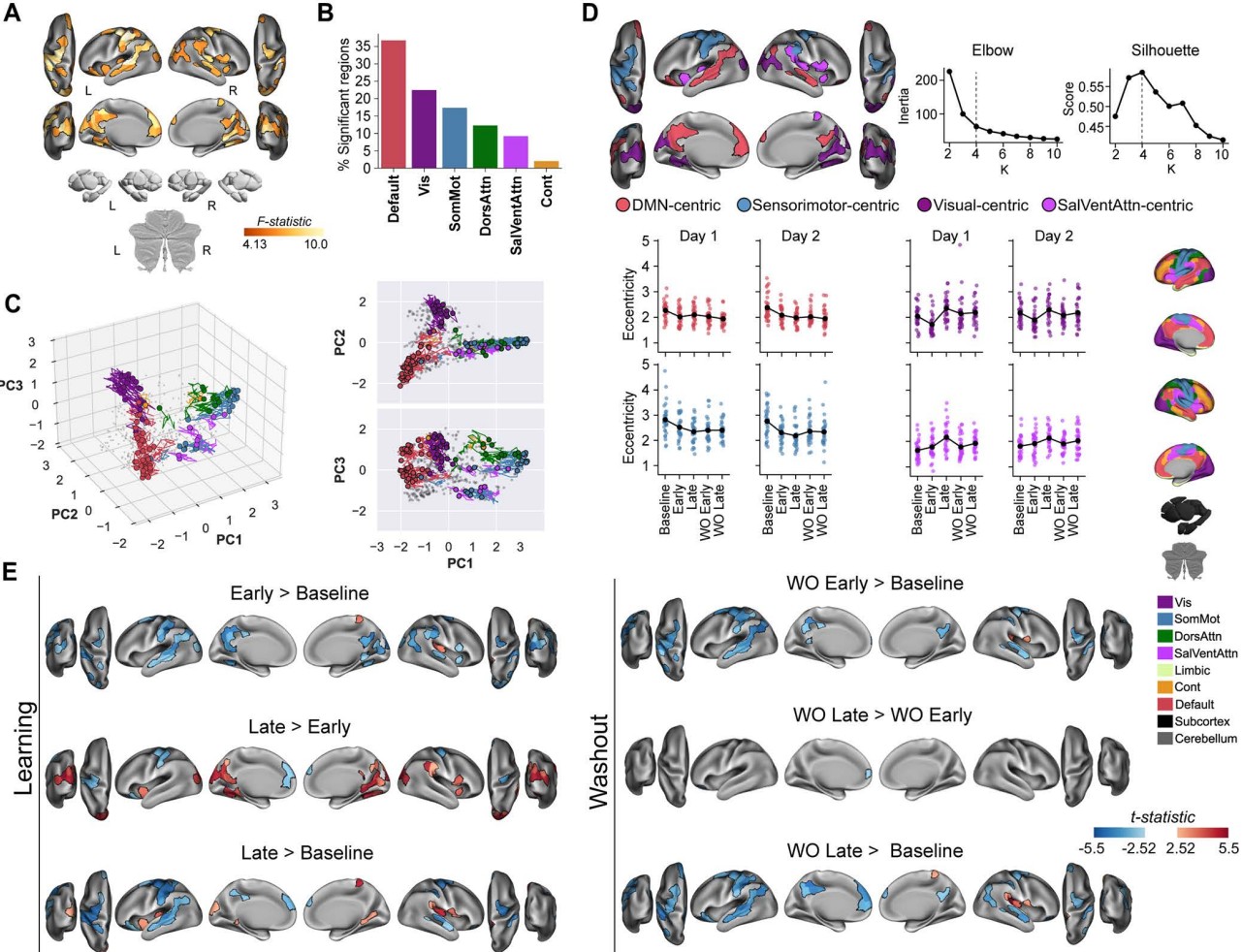

**Fig 3. Consistent task-epoch dependent changes in manifold eccentricity across days. (A)** Brain regions exhibiting a significant main effect of Task Epoch identified via a region-wise Day × Task Epoch rmANOVA (FDR corrected, $q < 0.05$). Color bar indicates F-statistic values. **(B)** Percentage of the significant regions from (A) across the seven canonical functional networks defined by Yeo et al. 7-network parcellation [89]. **(C)** Low-dimensional neural trajectories for significantly modulated regions from **A.** Circles mark the Day 1 Baseline position, with traces showing displacement across the subsequent nine epochs (Baseline, Early/Late Learning, Early/Late Washout for Day 1, then same sequence for Day 2). Gray points represent non-significant regions. **(D)** Four functional ensembles derived via k-means clustering of significant regions based on their Baseline coordinates. Brain maps show the spatial distribution of each ensemble. Scatter plots below show mean eccentricity per participant within each ensemble across epochs, with black markers indicating the group average. Note the consistent dynamics across Day 1 and Day 2. **(E)** Statistical maps showing results from pairwise t-tests comparing eccentricity between task epochs (averaged across Day 1 and Day 2; FDR corrected, $q < 0.05$). Red indicates significant manifold expansion (increased eccentricity; i.e., greater segregation); blue indicates significant manifold contraction (decreased eccentricity; i.e., greater integration). Underlying numerical data are provided in S1 Data and archived on Zenodo (https://doi.org/10.5281/zenodo.18613054).

PLOS Biology

connectivity-based changes were not related to simple changes in mean BOLD activation in these regions across epochs (S3 Fig).

To summarize these regional effects, we performed *k*-means clustering based on each significant region's embedding (i.e., geometric coordinates in PC space) within the Baseline manifold space. This analysis grouped the significantly modulated regions into four different functional ensembles: a DMN-centric ensemble, a Sensorimotor-centric ensemble, a Salience/Ventral Attention (SalVentAttn)-centric ensemble, and a Visual-centric ensemble (Fig 3D, brain maps). Consistent with different functional roles during learning and relearning, each ensemble exhibited dynamic manifold eccentricity changes that were both distinct from the other ensembles, and consistent across both days (Fig 3D, right panel).

Follow-up pairwise contrasts (averaged across days, FDR corrected $q < 0.05$) revealed these different dynamics across epochs (Fig 3E). Early learning (vs. Baseline) was characterized by significant manifold contraction (decreased eccentricity, increased integration) primarily within DMN-centric, Sensorimotor-centric and Visual-centric ensembles (regions in blue). In contrast, the transition to late learning (Late > Early) involved manifold expansion (increased segregation) in the Visual-centric ensemble and parts of the DMN (e.g., PCC, regions in red in Fig 3E), while primary motor cortex (M1) and DMN regions like mPFC now showed contraction (regions in blue in Fig 3E). Crucially, the contrast between late learning and baseline (Late > Baseline), representing states of comparable behavioral performance (minimal error, see Fig 1B), revealed that DMN-centric and Sensorimotor-centric ensembles remained significantly contracted relative to Baseline, while SalVentAttn-centric and Visual-centric ensembles remained significantly expanded.

Notably, we observed a similar set of dynamics during the washout phase. Specifically, we found that Washout mirrored Learning, in that both Early Washout and Early Learning were characterized by the contraction of DMN/Sensorimotor ensembles relative to Baseline, while both Late Washout and Late Learning maintained these contractions, but also showed expansion of SalVentAttn/Visual regions relative to Baseline (note that this similar set of dynamics was also clearly observable in the unthresholded stat maps, S4 Fig). Notably, this persistence of manifold changes during both Late Learning and Late Washout, even though the behavioral errors were minimal and similar to Baseline-levels, provides strong evidence for an enduring, latent neural change induced by initial adaptation. Moreover, the finding that there was no effect of Day or any interaction highlights that there was a strong similarity in how the brain reconfigured during both initial learning (Day 1) and subsequent relearning (Day 2). This strong similarity suggests that motor savings is underpinned by the re-expression of connectivity patterns established during Day 1 learning. In the next section, we will test this re-expression hypothesis more directly.

## Manifold patterns during early learning are re-expressed during early relearning

To test the hypothesis that faster relearning on Day 2 (savings) involves the re-expression of early learning neural patterns from Day 1, we turned to a method that has been used to establish reinstatement of activity patterns as fundamental to episodic memory retrieval: Representational Similarity Analysis (RSA; [92,93]). For this analysis, we compared the spatial pattern of *regional eccentricity values* during Day 1 Early Learning with the patterns from all subsequent epochs, separately for each functional ensemble. This approach conceptually parallels methods quantifying neural reinstatement in the episodic memory literature [29–38], while extending the approach from patterns of activity, to patterns of regional integration/segregation.

A specific reinstatement effect emerged within the DMN-centric ensemble (Fig 4). Here, we found that the pattern similarity (Pearson correlation, r) between Day 1 Early Learning and the other epochs significantly differed (rmANOVA: $F(6, 186)=2.26$, $p=0.040$), with the Day 2 Early Relearning epoch correlation being significantly higher than the correlations for nearly all other subsequent epochs (including the adjacent timepoints of Day 1 Late Learning and Day 2 Late Relearning, all $p$s < 0.031). This is consistent with the hypothesis that the connectivity patterns associated with early learning on Day 1 are uniquely re-expressed during early relearning on Day 2. In addition, we found that the Sensorimotor-centric ensemble also showed significant modulation across epochs ($F(6, 186)=2.54$, $p=0.022$), though it lacked this same unique

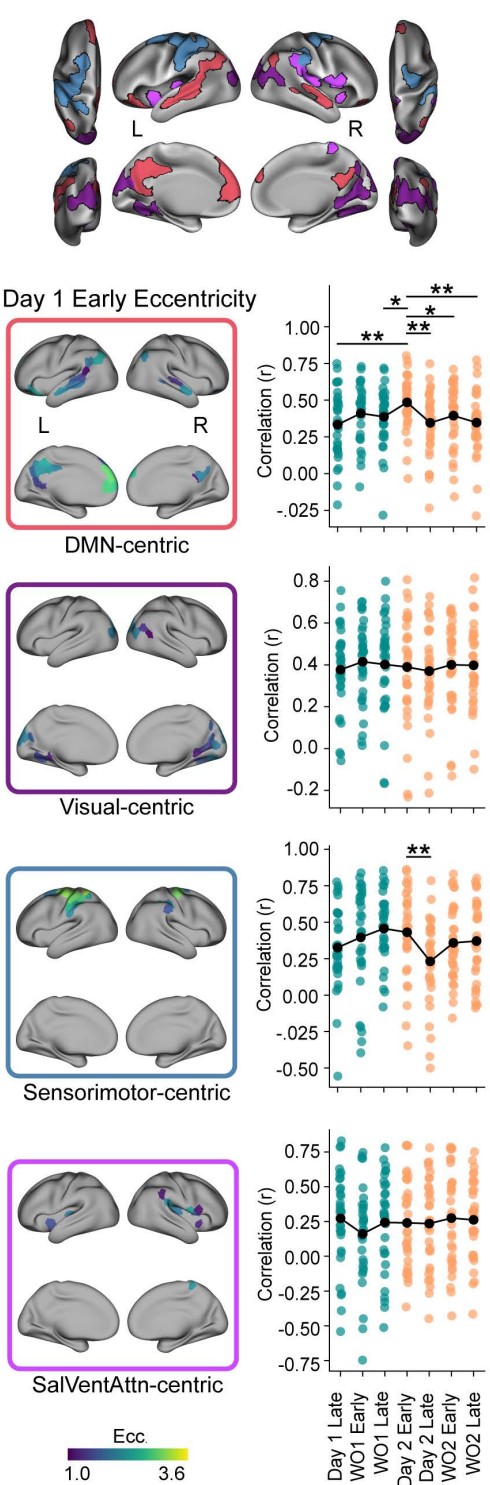

**Fig 4. Reinstatement of DMN-centric manifold structure during relearning.** Left column panels depict the mean regional eccentricity values during the Day 1 early learning epoch for each of the four functional ensembles identified in Fig 3 (and shown at top). Right column panels show the results of the Representational Similarity Analysis (RSA). Each plot displays the pattern similarity (Pearson correlation, r) between the Day 1 early learning epoch eccentricity values and the values during the subsequent learning and washout epochs across both days, calculated separately for each ensemble. The black line shows the average across subjects, while individual data points represent single subjects, color-coded by day (Day 1: cyan; Day 2: orange).

Following a significant main effect of epoch in the overall rmANOVA, asterisks denote significant post-hoc differences identified from one-tailed paired samples t-tests comparing the similarity between Day 1 Early Learning and Day 2 Early Relearning vs. other epochs (* $p < 0.05$, ** $p < 0.01$). Note the selective reinstatement effect (highest correlation with Day 2 Early Relearning) primarily within the DMN-centric ensemble (top right). Underlying numerical data are provided in S1 Data and archived on Zenodo (https://doi.org/10.5281/zenodo.18613054).

reinstatement signature specific to the early relearning phase; the only significant difference was a higher correlation between Day 1 Early Learning and Day 2 Early Relearning compared to Day 2 Late Relearning (Fig 4).

The selectivity of this DMN reinstatement effect—heightened similarity between Day 1 early learning and Day 2 early relearning—argues against explanations based on simple time-dependent factors (e.g., fatigue) or general neural drift, which would instead predict temporal similarities between adjacent epochs or less functional network-level specificity. Furthermore, the lack of an obvious reinstatement effect in the Visual-centric ensemble suggests this phenomenon is not purely driven by visual error signals (which are highest during the early learning and early washout phases on both days). Instead, the selective DMN reinstatement might suggest a role for this network in retrieving or re-implementing higher-order aspects of the learned behavior, such as recalling the learned context [23] or a cognitive re-aiming strategy [9,94], consistent with the DMN's established functions in both memory retrieval and strategic processing [54,56,95–97].

### Network interactions driving changes in manifold embedding

While changes in manifold eccentricity signify alterations in a region's overall functional embedding, the specific changes in inter-regional connectivity that drive these shifts remained to be determined. To elucidate the network interactions underlying the observed Task Epoch effects (Fig 3), we performed seed-based FC analyses. We selected representative regions from the functional ensembles that showed prominent task-epoch modulation, chosen to reflect the core functional anatomy of each ensemble: the left PCC and mPFC from the DMN-centric ensemble, the left primary motor cortex (M1) from the Sensorimotor-centric ensemble, and a region in left lateral visual cortex (Visual) from the Visual-centric ensemble. For each seed, we contrasted its whole-brain connectivity map between key task epochs, specifically examining changes from Baseline to Early learning/relearning (Early > Baseline) and from Early to Late learning/relearning (Late > Early), averaging these contrasts across both days (Fig 5A; see S5 Fig for right hemisphere homologs). As eccentricity is a multivariate index of a region's connectivity that takes into account patterns across the whole brain, we present the unthresholded seed-connectivity maps to provide a comprehensive view of these network changes.

The left PCC seed (DMN ensemble) exhibited a dynamic two-phase pattern: significant manifold contraction from Baseline to Early learning, followed by significant expansion from Early to Late learning (Fig 3E). These opposing shifts in manifold embedding were underpinned by a striking reconfiguration of its FC (Fig 5A, top row). The initial contraction phase (Early > Baseline) was driven by a marked decrease in connectivity with other core DMN regions (including the angular gyrus and middle temporal cortex), the hippocampus, and cerebellar association areas. Concurrently, the PCC increased its coupling with key sensorimotor areas, including the left primary motor cortex, superior parietal cortex, and premotor cortex. This connectivity profile then reversed during the later expansion phase (Late > Early). In this period, the PCC re-engaged with its own network, showing increased coupling with other DMN regions, while simultaneously decreasing its connectivity with the sensorimotor areas it had previously integrated with (note that these network reversals are nicely captured in the spiderplots in Fig 5C). This two-part dynamic—heightened sensorimotor coupling during early learning followed by a return to stronger intra-DMN coupling during late learning—provides a compelling mechanistic account for the observed manifold dynamics. It also directly supports our interpretation of manifold contraction as heightened between-network integration and manifold expansion as a return to within-network segregation.

In contrast, both the left mPFC (DMN ensemble) and the left M1 (Sensorimotor ensemble) showed significant manifold contraction as they transitioned from Early to Late learning (Fig 3E). For the mPFC, this later-phase contraction (Late > Early) was associated with increased connectivity with networks outside the DMN, including premotor cortex as

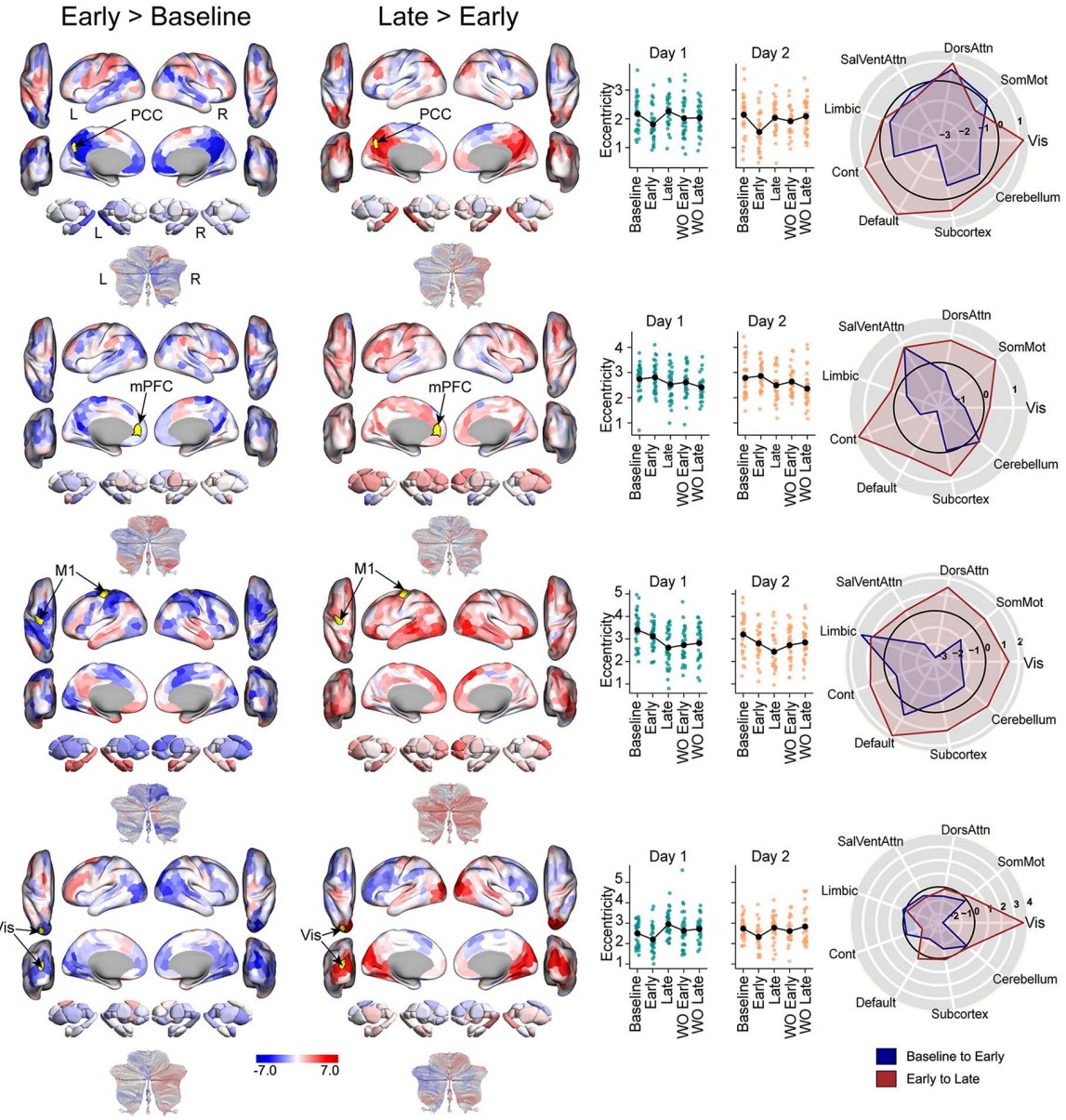

**Fig 5. Connectivity changes that underlie the Task Epoch effect.** This figure illustrates changes in whole-brain functional connectivity for representative seed regions (indicated in yellow and by arrows) selected based on the significant main effect of Task Epoch from the ANOVA (see Fig 3A). Each row corresponds to a specific seed region. **(A)** Brain maps display the results of paired t-tests contrasting connectivity patterns between different task epochs (averaged across Day 1 and Day 2). For a given contrast (e.g., Early > Baseline) red denotes increased connectivity in the first epoch relative to the second, while blue denotes decreased connectivity. Specific contrasts are shown above the maps. **(B)** Scatter plots show the manifold eccentricity trajectory for the corresponding seed region across all 10 task epochs. Each point represents the mean eccentricity for one participant in a given epoch, color-coded by Day (Day 1: cyan, Day 2: orange). The solid line overlay indicates the group mean, illustrating the consistency of eccentricity changes across both days. **(C)** Spider plots provide a concise, network-level summary of the whole-brain connectivity changes shown in panel **A**. Whereas the brain maps in (**A**) show hundreds of regional changes, the spider plots distill these effects by showing the average change in connectivity between the seed region and all regions within each of the seven canonical functional networks defined by the Yeo et al. 7-network parcellation [89]. The radial axis represents the t-statistic for this average change. Points outside the black circle ($t > 0$) indicate a net increase in connectivity between the seed and that network for the specified contrast (e.g., Early > Baseline), while points inside the circle ($t < 0$) indicate a net decrease. This allows for a clear and immediate visualization of how a seed region's network-level communication profile is reconfigured across task phases. Underlying numerical data are provided in S1 Data and archived on Zenodo (https://doi.org/10.5281/zenodo.18613054).

well as areas of the SalVentAttn (Fig 5A, second row). This suggests that as learning progresses, the mPFC—a DMN node typically more insulated from sensory inputs [65]—increasingly integrates with action-oriented brain networks. Similarly, M1's late-phase contraction (Late > Early) was driven by increased connectivity not only within the broader sensorimotor system but also with disparate networks like the DMN (Fig 5A, third row). In both instances, the observed manifold contraction corresponds directly with increased between-network functional coupling.

Finally, the left lateral visual cortex displayed a manifold profile similar to the PCC, contracting from Baseline to Early learning before expanding from Early to Late learning (Fig 3E). The initial contraction (Early > Baseline) was associated with decreased connectivity within the visual network itself, coupled with increased connectivity to regions outside its network, particularly in the left prefrontal cortex (Fig 5A, bottom row, left map). This fits the model of early integration reflecting enhanced between-network coupling. Conversely, the subsequent expansion (Late > Early) was driven by a clear reversal of this pattern: visual cortex now showed strongly increased within-network connectivity alongside decreased connectivity to the prefrontal regions it had engaged earlier (Fig 5A, bottom row, right map). This later phase, characterized by strengthening intra-network communication, aligns with the interpretation that manifold expansion reflects increased network segregation.

Critically, reinforcing the overarching main effects on eccentricity (Fig 3E), highly similar patterns of connectivity changes for these same seed regions were also observed during the corresponding washout phases, albeit relatively more muted (e.g., the Early Washout > Baseline contrast for the PCC mirrored its Early Learning > Baseline contrast; S6 Fig). This underscores the strong parallel between the large-scale network dynamics engaged during learning/relearning and those active during subsequent washout periods. In summary, these seed-based analyses provide concrete examples illustrating how specific changes in FC—primarily shifts towards increased between-network coupling (integration) for manifold contraction and increased within-network coupling (segregation) for manifold expansion—drive the observed alterations in regional manifold eccentricity.

## Changes in manifold structure relate to individual differences in learning and relearning

While our group analyses reveal the general trends in adaptation and savings (Fig 1B), it is well recognized that substantial inter-individual variability exists in motor learning [94,98–100]. This was also evident in our data, with some participants showing rapid adaptation and de-adaptation across both testing days, whereas others learned and washed out much more slowly (Fig 6A, example subjects). Crucially, a subject's performance in one phase was highly predictive of their performance in others. For instance, initial errors during early learning on Day 1 were strongly correlated with initial errors during early relearning on Day 2 (averaged over a 48-trial window/six 8-trial bins; $r = 0.74$, Fig 6B), and a similar stability was observed for early washout performance across days (averaged over a 48-trial window/six 8-trial bins; $r = 0.79$; Fig 6C). This high degree of intra-subject stability suggests that a subject's learning proficiency is a stable trait that manifests across the entire experimental timeline, rather than a transient state captured by a single learning window. Consequently, focusing on isolated metrics like initial error or late adaptation rate would fail to capture this overarching, continuous aspect of individual performance.

To distill this stable, subject-wide variation into a single comprehensive metric, we therefore applied functional principal component analysis (fPCA) [101] to each participant's complete two-day learning curve (see Materials and methods for further details; see also [74,85]). This data-driven approach is ideally suited for this purpose, as it identifies the dominant modes of variation in subjects' adaptation trajectories over time without being restricted to pre-defined epochs. The first component, capturing the majority (63%) of the variance, clearly represented subjects' overall learning proficiency across the entire experiment (Fig 6D), including both adaptation and de-adaptation phases. As such, we termed each subjects' loading onto this component as their "Learning Score" (Fig 6D, inset). Individuals with higher Learning Scores consistently adapted and de-adapted faster than the group average across both days (e.g., Fig 6A, green trace versus red trace), confirming this score as a robust and holistic index of overall task performance (Fig 6E).

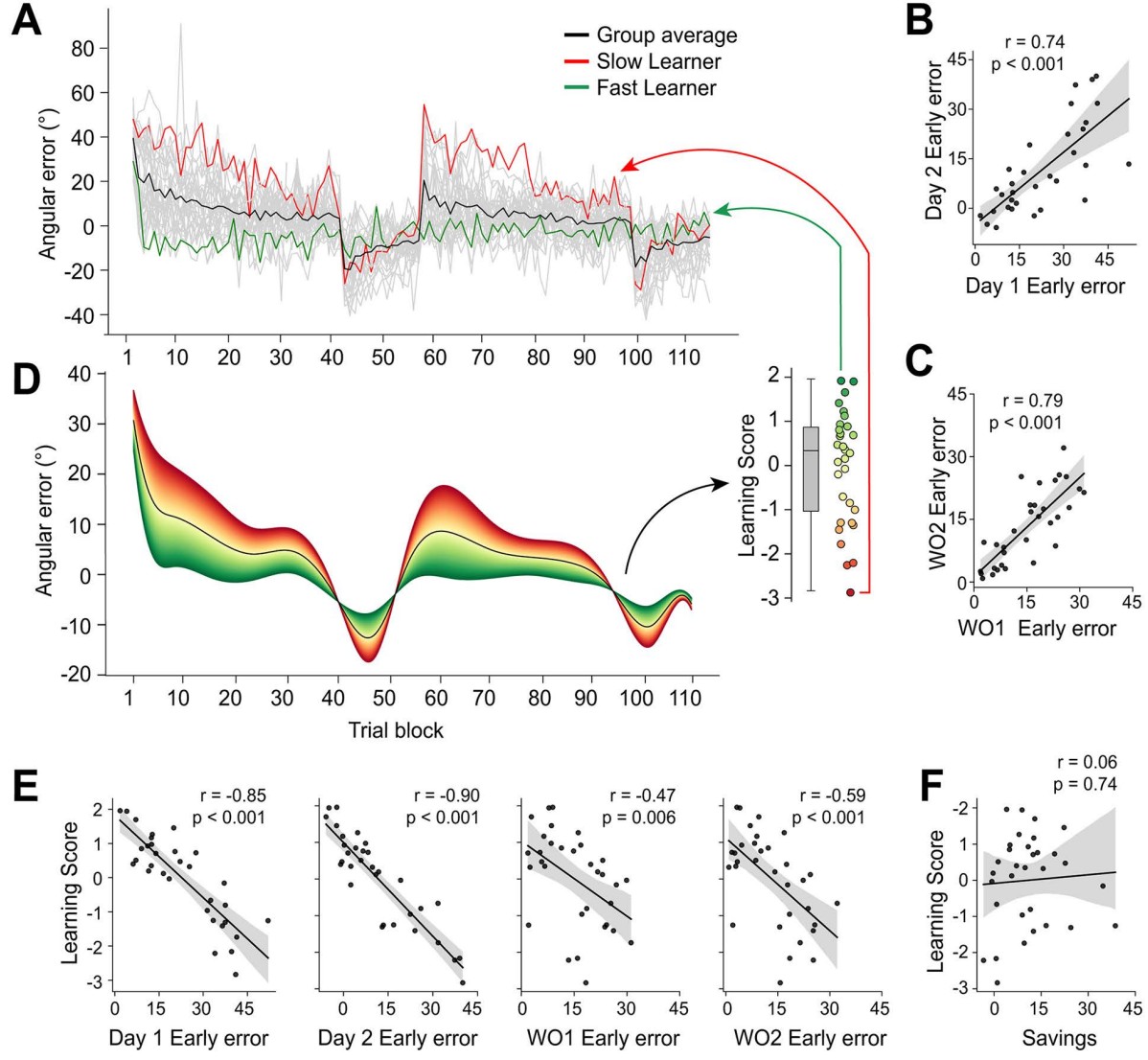

**Fig 6. Characterizing individual differences in learning performance. (A)** Learning curves (angular error across 8-trial bins) for all individual participants (light gray traces), overlaid with the group mean (black). Example curves for a "fast learner" (green) and a "slow learner" (red) are highlighted. **(B)** Correlation between participants' median error during early learning on Day 1 and early relearning on Day 2. **(C)** Correlation between participants' median error during early washout on Day 1 and early re-washout on Day 2. **(D)** Functional principal component analysis (fPCA) identified the dominant mode of variation across individual learning curves. Left: The first component ("Learning Score") captured 63% of the variance. Green and red shaded areas illustrate the effect of positive (faster learning) and negative (slower learning) scores relative to the mean performance curve (black). Note that subjects exhibiting the highest and lowest Learning Scores are denoted in green and red, respectively, in panel **A**. Right inset: Distribution of individual participant loadings (scores) on this component. **(E)** Correlations between the derived Learning Score and median error during specific task epochs (Day 1 Early Learning, Day 2 Early Relearning, Day 1 Early Washout, Day 2 Early Re-Washout). Higher Learning Scores correlate with lower errors (faster adaptation/de-adaptation). **(F)** No significant correlation was found between behavioral savings (Day 1 initial error - Day 2 initial error) and the overall Learning Score. For scatter plots (**B**, **C**, **E**, **F**), each point represents one participant, the black line is the best-fit regression line, and shading indicates the 95% confidence interval.

We next related this overall Learning Score to behavioral savings, calculated as the difference in initial error between Day 1 and Day 2 (Day 1 initial error - Day 2 initial error). Notably, despite clear savings at the group level (Fig 1B), we found no significant correlation between individual participants' Learning Scores and the magnitude of their behavioral

savings ($r = 0.06$; Fig 6F). This suggests that while savings is an indication of motor memory retention, the magnitude of this effect at the individual level is complex and not actually related to subjects' overall learning performance, as captured by our fPCA measure. We also examined the "Recall Ratio" (RR) [102]—a behavioral metric quantifying the extent to which early relearning matches late day 1 learning ($RR = Adaptation_{D2\ Early} / Adaptation_{D1\ Late}$). We found that this ratio strongly correlated with our Learning Score ($r = 0.76$, $p < 0.001$), confirming that our fPCA-derived metric effectively captures proficiency in recalling the learned state (S9A–S9B Fig).

Having established the Learning Score as a robust measure of individual differences in overall adaptation and de-adaptation across days, we next investigated its neural basis. We reasoned that the most pronounced neural changes related to learning proficiency would be evident during the most dynamic phases of adaptation. Therefore, we focused our initial analysis by correlating the Learning Score with the change in manifold eccentricity during the critical early learning phases (Early > Baseline contrast) for Day 1 and Day 2 separately. While few individual brain regions survived a strict FDR correction (see Fig 7A for Day 1 example), the resulting correlation maps revealed a striking degree of spatial contiguity. Rather than being randomly scattered, the strongest correlations clustered neuroanatomically, specifically within trans-modal association cortices and the DMN-centric cluster in particular, like the PCC and mPFC. This non-random, structured pattern strongly motivated a shift from a region-wise to a network-level analysis [103,104]. We therefore examined correlations between the overall Learning Score and the average change in eccentricity within canonical functional networks [89], a more powerful approach for detecting distributed effects. The statistical significance of these network-level correlations was assessed using spin-test permutations (FDR corrected, $q < 0.05$; [103,104] see Materials and methods).

This network-level analysis revealed distinct relationships between network dynamics and overall learning performance that shifted across the two days. On Day 1, a higher Learning Score (better overall performance across both days) was significantly associated with greater contraction (negative correlation) of the DMN and DAN during the initial learning phase (Fig 7A). Conversely, better performance was also linked to less contraction (positive correlation) within the Soma-tomotor network (Fig 7A). The correlations within the DMN and DAN suggest that individuals whose networks became more functionally integrated during early learning tended to show better overall performance across the two days, echoing the group-level contraction patterns that we observed during early learning (Fig 3E). Notably, similar whole-brain correlation patterns between network eccentricity changes and Learning Score were observed during the early washout phase on Day 1 (S7 Fig), further highlighting parallels between learning and washout network dynamics. In an exploratory analysis relating neural eccentricity during late learning (Day 1) to the RR, we observed a widespread trend toward negative correlations across the cortex, indicating that better recall is associated with global manifold contraction (S9C Fig). However, unlike the Learning Score results, this effect was not selective; no single functional network survived permutation testing for network-specific effects. This suggests that while strategic recall involves broad cortical integration, the stable individual differences captured by our Learning Score are more robustly linked to specific network-level dynamics.

On Day 2, however, the relationship between DMN dynamics and performance reversed. Better overall performance (higher Learning Score) was now associated with a greater manifold expansion (positive correlation) of the DMN, and also the Frontoparietal Control network (Control), during the early relearning phase (Fig 7B). This switch—from DMN contraction correlating with better performance on Day 1 to DMN expansion correlating with better performance on Day 2—points to a fundamental shift in this network's role or state between initial learning and memory-guided relearning. Based on the brain maps, the Day 1 effect seemed weighted towards posterior DMN regions (e.g., PCC), while the Day 2 effect involved more anterior prefrontal DMN regions (e.g., mPFC). Again, mirroring the learning phase, we also observed a very similar spatial pattern of whole-brain correlations during the early re-washout phase on Day 2 (S7 Fig).

To understand the network interactions driving these performance-related eccentricity changes, particularly the reversal observed for the DMN, we examined how subjects' Learning Scores correlated with changes in the DMN's FC with other brain networks during the early learning phases of Day 1 and 2. As with the seed-based results in Fig 5, we have opted to show the unthresholded correlation maps depicting these various relationships.

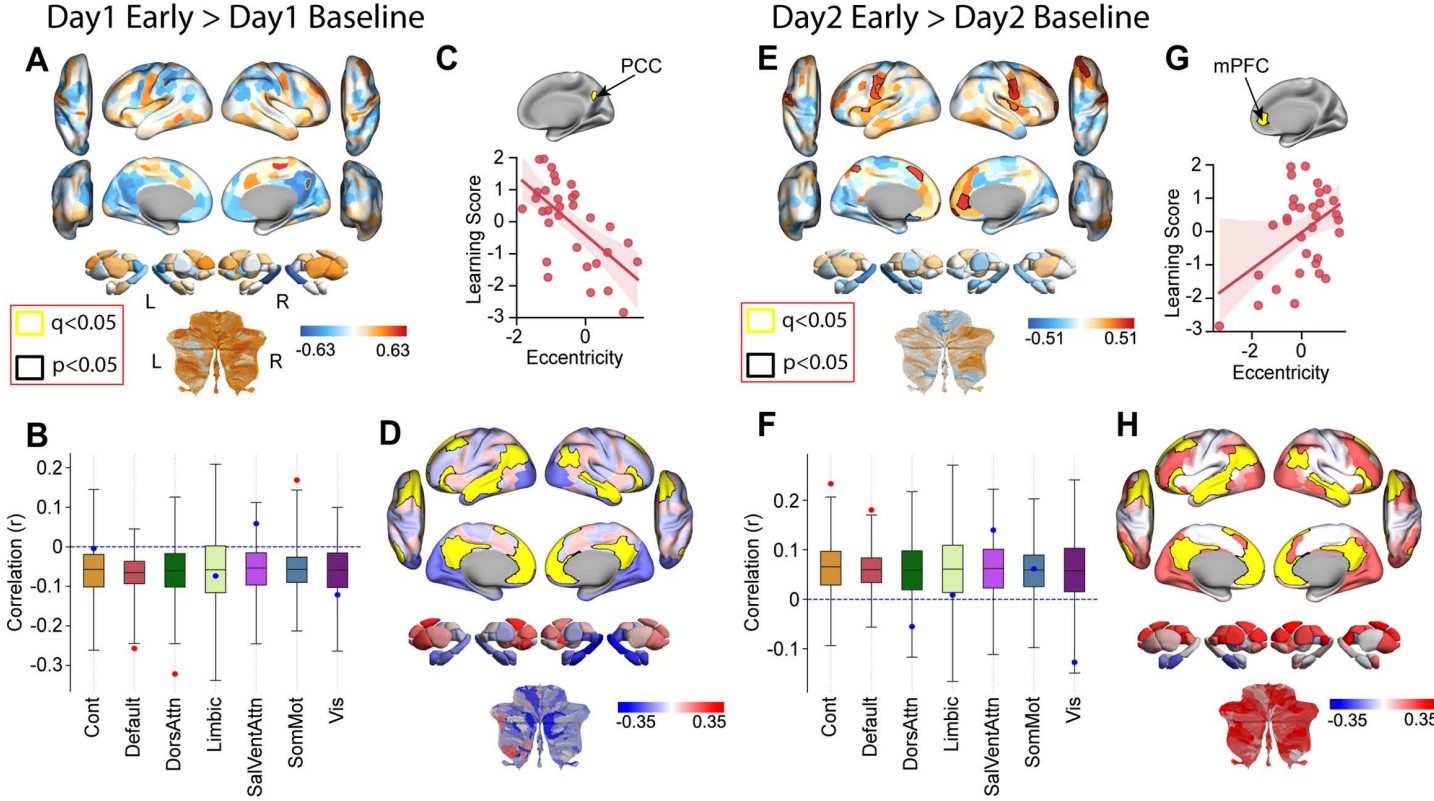

**Fig 7. Network eccentricity and connectivity changes correlating with individual learning performance. (A)** Whole-brain map displaying correlations between participants' Learning Scores and regional changes in manifold eccentricity during initial learning (Day 1 Early Learning > Day 1 Baseline). **(B)** To assess network-level significance, we used a spatial permutation "spin test." The analysis reveals which of the seven canonical networks show a significant correlation between Day 1 eccentricity changes (from A) and the Learning Score. Each data point shows the actual correlation for one of 7 Yeo et al. 7-network parcellation [89] networks. Boxplots depict the null correlation distribution (1000 iterations) for each network [103,104]. Boxplot elements: center line = median; box edges = 25th/75th quartiles; whiskers = min-max of null distribution. Dashed blue line indicates $r = 0$. The data points in red denote significant network-specific correlations (FDR corrected, $q < 0.05$). **(C)** Scatterplot illustrating the correlation from panel (A) for an example brain region (indicated in yellow) with participants' Learning Scores on Day 1. **(D)** Brain map illustrating how Day 1 changes in functional connectivity between the DMN and all other brain regions correlate with subjects' Learning Score. This map shows how the strength of DMN connectivity change correlates with the Learning Score. Red indicates that stronger DMN coupling with a given region is associated with better performance (higher Learning Score). Blue indicates that stronger DMN coupling is associated with worse performance (lower Learning Score). **(E)** Whole-brain correlation map as in **A**, but for eccentricity changes during early relearning on Day 2 (Day 2 Early Relearning > Day 2 Baseline). **(F)** Network-level spin-test results as in **B**, but for the Day 2 correlations shown in **E**. **(G)** Scatterplot illustrating the correlation from panel (E) for an example brain region on Day 2. **(H)** Brain map as in **D**, but showing how Day 2 changes in DMN connectivity correlate with the Learning Score. (S7 Fig presents corresponding analyses for washout periods; S8 Fig presents the brain maps in panels D and H for the other significant brain networks flagged by the spin-test procedure). Underlying numerical data are provided in S1 Data and archived on Zenodo (https://doi.org/10.5281/zenodo.18613054).

On Day 1, individuals with better overall performance (higher Learning Score) displayed a distinct DMN connectivity profile characterized by stronger functional coupling with subcortical structures (caudate, putamen), but notably weaker coupling with visual cortex, Control network, and large portions of the cerebellum (Fig 7D). This pattern of network interactions was presumably driven by posterior DMN nodes such as the PCC. Indeed, this region not only features prominently in the whole-brain correlation map (Fig 7A and 7C) but also itself exhibits significant contraction during early learning (Fig 5B, first row). PCC's contraction likely explains the greater contraction in the DMN more broadly observed in high performers (i.e., negative correlation with Learning Score) during initial learning.

On Day 2, a remarkably different DMN connectivity profile correlated with better overall performance. While the positive DMN-subcortical coupling remained, high performers now exhibited significantly stronger DMN coupling with the cerebellum, visual cortex, and the Control network—representing a clear reversal of the Day 1 pattern, particularly for the cerebellar and visual cortical interactions (Fig 7H). This altered pattern of network interactions was presumably driven by anterior DMN nodes such as the mPFC (Fig 7G). Indeed, this region not only features prominently in the whole-brain correlation map (Fig 7E) but also itself tended towards expansion in early learning (Fig 5B, second row). In contrast with the PCC's contraction in the posterior DMN during early learning, the mPFC's expansion likely explains the greater DMN expansion observed in high performers (i.e., positive correlation with Learning Score) during early relearning.

Thus, taken together, our findings reveal a multi-layered role for the DMN in motor learning. On one hand, our group-level analyses indicate substantial overlap and re-expression of task-epoch dynamics across days (Fig 3), particularly the DMN connectivity states that emerge during early adaptation and re-adaptation (Fig 4). This suggests a core, consistent mechanism of DMN engagement that underpins the learning and savings process for all individuals. Layered on top of this, however, our findings related to individual differences reveal a more nuanced picture. We found that distinct, performance-related DMN connectivity states are engaged during initial adaptation (Day 1 contraction) versus subsequent memory-guided relearning (Day 2 expansion). Crucially, these performance-related modulations were not simply stronger (or weaker) versions of the main group-level effects, but rather represented a more subtle, individualized tuning of network interactions that was related to differences in learning performance. This reversal highlights the DMN's functional flexibility, potentially leveraging different subsystems—such as posterior DMN (e.g., PCC) for initial learning versus anterior DMN (e.g., mPFC) for retrieval [65,105]. Together, this suggests that against the backdrop of a robust and largely consistent group-level reinstatement of the DMN's overall network state on Day 2, the DMN also slightly reconfigures its performance-critical connectivity to meet the distinct computational demands of forming new motor memories versus efficiently retrieving them.

## Learner heterogeneity reveals different DMN-centric reinstatement profiles in fast versus slow learners

To address the potential influence of distinct learning strategies on our findings, we considered evidence that rapid motor adaptation is often associated with greater reliance on explicit re-aiming processes, whereas slower adaptation may reflect a relatively larger contribution of implicit recalibration mechanisms [94,99,106,107]. Because our fPCA-derived Learning Score captures variance in learning profiles across the experiment (see Fig 6), it can serve as a proxy for stratifying participants by adaptation dynamics. We therefore stratified participants into "fast" and "slow" learners based on a median split of their Learning Score (Fig 8A). While we interpret the resulting patterns with caution given the reduced effective sample size ($n = 16$ per group), this stratification allows us to investigate whether the DMN reinstatement effect is uniform across the population or preferentially expressed in those exhibiting a fast learning profile.

A clear subgroup-dependent reinstatement effect emerged within the DMN-centric ensemble (Cluster 1; Fig 8B). In the fast learner subgroup, the DMN exhibited a clear reinstatement profile: pattern similarity during Day 2 Early Relearning was significantly higher than in subsequent epochs (including the adjacent timepoints of Day 1 Late Learning and Day 2 Late Relearning; all $p$s < 0.05). This reproduces the unique DMN reinstatement signature observed in the full sample, indicating that connectivity patterns associated with initial learning are preferentially re-expressed during early relearning in participants with a fast learning profile.

In contrast, the slow learner subgroup exhibited a markedly more flattened profile. Specifically, we observed no significant elevation of similarity during Day 2 Early Relearning relative to other epochs (all $p$s > 0.05), with the exception of the Day 2 Late Relearning epoch. Notably, this muted reinstatement effect in slow learners occurred despite robust behavioral savings being present across participants, further suggesting that the DMN effect observed in fast learners is not driven by generic error signals (which are present during early relearning across both subgroups). Together, these results suggest that the DMN-centric reinstatement signature is preferentially expressed in participants with a fast learning profile,

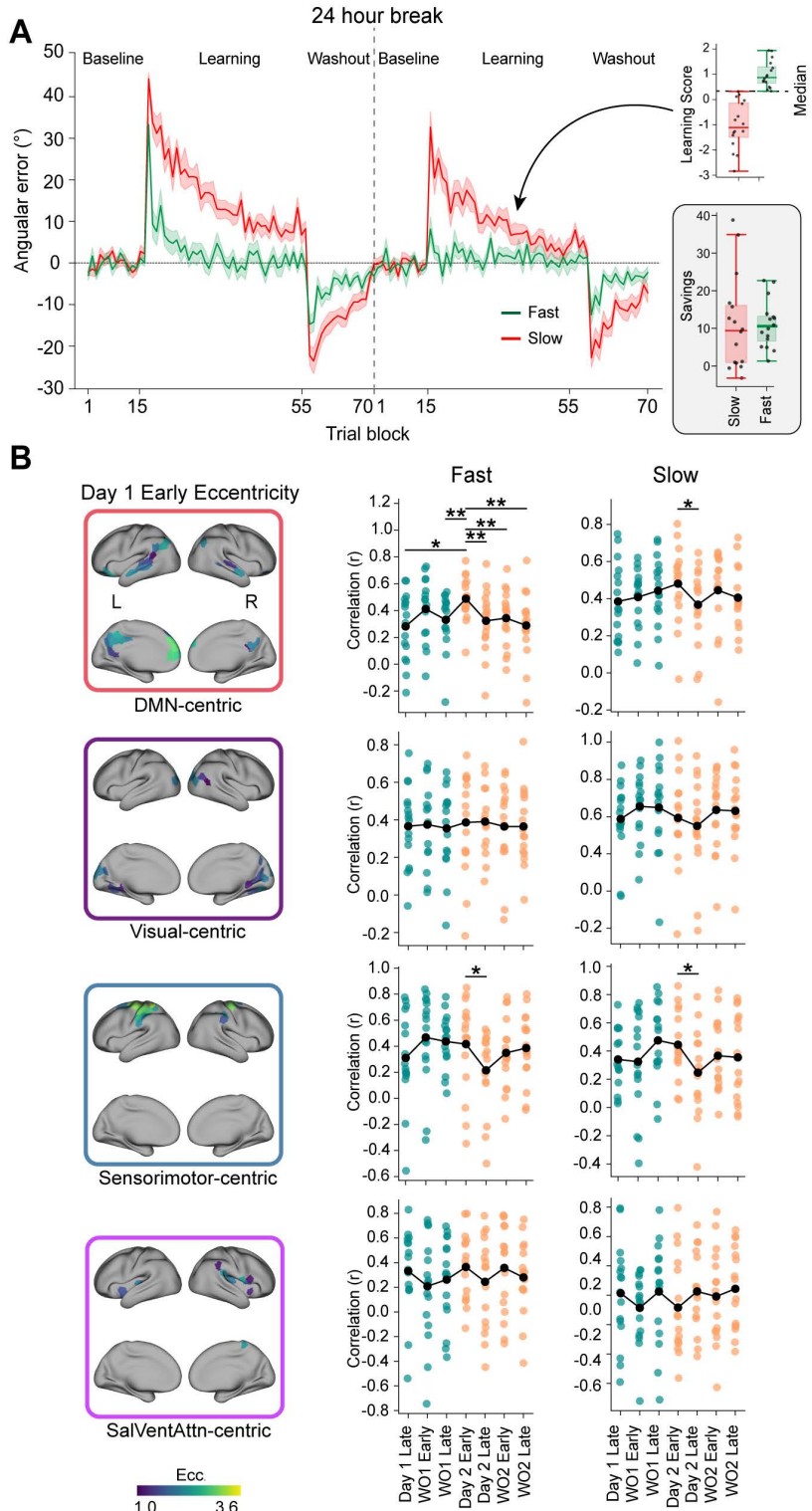

**Fig 8. Learner heterogeneity reveals different DMN-centric reinstatement profiles in fast vs. slow learners. (A)** Behavioral learning curves (angular error across trial blocks; 8-trial bins) for participants split into fast (green) and slow (red) learners via a median split on Learning Score (top right inset). The timeline includes Baseline, Learning, and Washout phases on Day 1 and Day 2 separated by a 24-hour break. The right panel depicts the

median-split subgrouping: Learning Score (top) and savings (bottom) for slow vs. fast learners, shown as boxplots with individual participant data points. **(B)** Subgroup reinstatement/similarity analysis. Left panels show Day 1 Early eccentricity maps for each ensemble (DMN-centric, Visual-centric, Sensorimotor-centric, SalVentAttn-centric) identified in Fig 3. Right column panels show the results of the representational similarity analysis (RSA) plotted separately for fast and slow learners. Each plot displays the pattern similarity (Pearson correlation, r) between the Day 1 Early learning epoch eccentricity values and the values during the subsequent learning and washout epochs across both days, calculated separately for each ensemble. The black line shows the average across subjects, while individual data points represent single subjects, color-coded by day (Day 1: cyan; Day 2: orange). Asterisks denote significant differences identified from one-tailed paired samples t-tests comparing the similarity between Day 1 Early Learning and Day 2 Early Relearning vs. other epochs (* $p < 0.05$, ** $p < 0.01$). Note the selective reinstatement effect (highest correlation with Day 2 Early Relearning) primarily within the DMN-centric ensemble (top right), which is magnified in fast learners. Underlying numerical data are provided in S1 Data and archived on Zenodo (https://doi.org/10.5281/zenodo.18613054).

consistent with differential engagement of retrieval-related or strategy-related processes, even when the magnitude of behavioral savings is comparable across groups.

## Discussion

A hallmark of successful behavior is the brain's capacity to swiftly reacquire previously learned motor skills. This effect, termed "savings," suggests the presence of an enduring memory trace associated with the initial learning experience. Yet, how and where in the brain these traces persist, and how they are retrieved to guide future action, remains poorly understood. Our study, using human fMRI and whole-brain manifold learning across two days of visuomotor adaptation, indicates that motor savings is underpinned by the reinstatement of large-scale cortical FC patterns (manifold structures) established during initial learning. Importantly, we found that this neural reinstatement was not diffusely observed across sensorimotor systems but was predominantly localized to regions within the DMN, a distributed set of brain regions mainly implicated in memory-guided cognition [45,48,56,95]. This finding is particularly profound when considered alongside our recent work showing that the DMN also underpins the transfer (or generalization) of motor learning [63]. Together, these results establish a central and unifying role for the DMN in the flexible deployment of motor memories, using a common mechanism—the reinstatement of context-appropriate network states—to guide behavior both across time (savings) and across contexts (transfer). This finding, paralleling reinstatement principles from other memory domains (episodic memory, fear conditioning) and anticipated by recent computational models of motor learning [23], suggests a common mechanism for the flexible recall and reuse of stored memories across diverse behavioral contexts.

As noted, the prominent role of the DMN in motor memory retrieval is particularly compelling, given its traditional association with internally-directed cognition, such as episodic memory and introspection, rather than direct motor control [45,47,48,56,108]. However, the DMN's established functions in retrieving contextual details [45–48], integrating information across distributed brain systems [56,64,65,109], and supporting strategic, goal-directed thought [54,56,97], make it a strong candidate for orchestrating the recall of learned motor adaptations. The DMN's unique neuroanatomical position at the apex of the cortical processing hierarchy [64,65] may allow it to reinstate the broader cognitive and contextual state associated with initial learning, thereby facilitating the re-engagement of appropriate motor programs [3,22] or associated cognitive strategies [9,24,25]. This interpretation aligns with recent influential computational frameworks, such as the contextual-inference model, which posit that savings is driven by the re-expression of context-dependent memory traces [23], and is consistent with our recent work implicating the DMN in the strategic transfer of motor learning across hands [63].

The persistence of learned information, even when no longer manifested behaviorally, has long suggested the existence of latent "memory traces" in the brain [7,14,110]. In recent years, these classic ideas have received considerable support from both animal and human work. For instance, neurophysiological studies from non-human primates and rodents have demonstrated long-lasting, learning-induced memory traces within the motor cortex [26–28]. Furthermore, fear conditioning studies in rodents show that specific neuronal ensembles activated during learning (engrams) are reactivated upon re-exposure to the learned context [15–17,39]. This principle extends to humans, where episodic learning studies have shown that multivoxel patterns of brain activity in the DMN and visual cortex, present during learning, are

reinstated at retrieval, facilitating successful memory [29–42], paralleling our observation that network-level reinstatement is most prominent in participants with stronger savings.

Our observation that DMN manifold structures from initial learning are re-expressed during early relearning provides a human network-level correlate of the "memory engram" concept, extending it to the domain of motor learning. One might argue that this pattern similarity does not reflect a specific memory trace, but rather a more general neural state associated with high error or cognitive effort, as both early learning and early relearning are among the most demanding task phases. However, this interpretation as a simple error-monitoring signal is unlikely for three reasons. First, the reinstatement effect was selective to the DMN-centric ensemble, rather than being broadly observed across other cognitive control and attention networks that are more commonly associated with being sensitive to task demands and error monitoring [111–113]. Second, the reinstatement profile appeared to track learning proficiency rather than error magnitude: the selective elevation of Day 2 Early similarity was most clearly expressed in fast learners. In contrast, slow learners—who experience prolonged error signals—did not cleanly show a corresponding reinstatement peak. This dissociation is difficult to reconcile with a generic "high error" or "high effort" account. Finally, and perhaps most importantly, both our late learning and late washout results show a persistence of learning-induced DMN manifold changes during periods of minimal (baseline-level) motor error (Fig 3E), further arguing against an explanation based on ongoing error processing. Instead, the evidence points toward the reactivation of a specific network state established during initial learning that can be selectively re-engaged to support subsequent relearning.

Crucially, our emphasis on the DMN does not undermine the critical role of the motor cortex in motor memory formation and retrieval [26–28]. Indeed, the distinction between our results and prior neurophysiological findings likely reflects different scales of analysis. Whereas previous work has selectively focused on the neural population activity within motor cortical areas [26–28], our whole-brain manifold approach captures global shifts in functional organization. From this broader perspective, we can highlight the DMN's potential role in orchestrating the large-scale reactivation of these more specific, cortically-held motor engrams.

Our seed-based connectivity analyses offer direct insight into how the DMN might fulfill this orchestrating role. We found increased functional coupling between the PCC—a key DMN node implicated in contextual processing and strategy formation [54,56,97]—and sensorimotor regions during both early learning and relearning (Fig 5). This heightened integration, with both DMN and sensorimotor regions exhibiting manifold contraction during these phases (Fig 3E), suggests a hierarchical mechanism for the DMN to reinstate the broader cognitive and contextual state associated with initial learning. Under this view, the DMN acts as a higher-level system that, upon re-exposure to a learned context, triggers the re-expression of motor memories within sensorimotor cortices. The subsequent decoupling between the PCC and sensorimotor cortex during late learning, once the context is well-instantiated, further supports this dynamic, context-dependent role. While fMRI precludes definitive claims of causality, this pattern of integration followed by decoupling is highly consistent with a hierarchical model where the DMN provides a contextual signal that gates or informs sensorimotor processing.

The dynamics of the manifold changes observed during the washout phase further reinforces the concept of a persistent, learning-induced memory trace. Specifically, our observation that the early washout phases mirrored the early learning phases in terms of DMN and sensorimotor manifold changes (Fig 3E) suggests that the learned adaptation does not simply decay or is erased [26–28]. Instead, it may revert to, or actively engage, a neural state that monitors ongoing errors and adjusts motor output accordingly, a process that itself shows strong similarities to initial learning [6]. More strikingly, the failure of many DMN and sensorimotor regions to fully revert back to their pre-learning, baseline positions on the manifold, even during late washout (Fig 3E), indicates an enduring alteration in functional organization—a potential neural signature of the latent memory that supports savings. This persistence of learning-induced connectivity changes echoes Semon's original engram concept: A dormant state awaiting "ecphory", the re-awakening by appropriate retrieval cues [14,110,114].

While our group-level RSA results highlight the *reinstatement* of the overall DMN manifold state during relearning, our individual differences analysis revealed an important *reconfiguration* of the specific DMN dynamics that predicted learning success. To our surprise, individual differences in DMN dynamics correlated with learning proficiency in distinct ways across days. On Day 1, greater DMN contraction (integration) during early learning predicted better overall performance. Conversely, on Day 2, greater DMN expansion (segregation) during early relearning predicted better overall performance (Fig 7). This reversal suggests a functional shift in the DMN's role between initial learning and memory-guided relearning, at least with respect to learning performance. As noted above, initial learning might necessitate broad DMN-orchestrated integration, perhaps reflecting active strategy formation and contextual binding. This aligns with the established role of the posterior DMN (including PCC, a main driver of our Day 1 effects) in adapting behavior and monitoring outcomes [54,55,97], regions shown to possess specific cytoarchitectural profiles well suited for integrating diverse inputs [65]. In contrast, a more refined DMN state focused on efficiently recalling and implementing an established strategy appears optimal for relearning. This could engage more anterior DMN subsystems (including mPFC, a main driver of our Day 2 effects), crucial for retrieving consolidated memories [115,116]. This functional toggling between network states for novel encoding versus memory-guided retrieval mirrors similar state-dependent dynamics observed in the hippocampus, where distinct connectivity patterns predict whether new information is being learned or existing memories are being successfully retrieved [117,118]. This anatomical heterogeneity within the DMN—characterized by a different, more "insulated" microarchitecture in anterior versus posterior nodes—provides a plausible substrate for the observed functional shift, allowing its different sub-systems to be preferentially engaged depending on whether the task demands novel learning or memory-guided retrieval [65]. Thus, the most parsimonious explanation for this dynamic reversal is that the DMN flexibly shifts between distinct functional states optimized for either memory encoding and strategy formation (integration-heavy) or for memory retrieval and application (segregation-heavy).

A critical consideration for interpreting these findings is the known heterogeneity of learning strategies within VMR tasks. As noted by prior work, individuals differ substantially in their reliance on explicit, strategic re-aiming versus implicit, error-based recalibration [9,43,94,106], leading to distinct learning profiles. To address this, we stratified participants into "fast" and "slow" learners—a common proxy for the relative contribution of explicit strategies [61,94,99]. We found that the DMN-centric reinstatement signature was much more strongly expressed in fast learners. In contrast, while slow learners exhibited a qualitative trend toward reinstatement, this effect was much more attenuated (Fig 8). This learner-dependent expression suggests that DMN recruitment may be preferentially associated with the rapid, memory-guided component of adaptation typical of strategic relearning, rather than the slower, incremental processes associated with implicit recalibration. However, we recognize that future work using paradigms designed to isolate implicit and explicit processes, such as error clamps [119–125] or delayed feedback [121,126], will be crucial for disentangling the DMN's precise contribution to each component of motor memory and for determining whether DMN reinstatement preferentially tracks one component over the other.

The dissociation between subjects' overall learning proficiency (Learning Score) and the magnitude of behavioral savings also becomes clearer when considering the contribution of different learning strategies. While seemingly counterintuitive, this decoupling likely stems from the outsized role of explicit cognitive strategies in VMR tasks [9,43]. As discussed, individuals can vary significantly in their reliance on these strategies. Those who rapidly develop and implement an effective re-aiming strategy on Day 1 may quickly reach asymptotic performance. This leaves limited room for further improvement—and thus measurable behavioral savings—on Day 2, regardless of their underlying capacity for implicit learning [85, see also 100]. This complex interplay between initial strategic performance, ceiling effects, and the efficiency of strategy retrieval can therefore obscure a simple relationship between overall learning ability and the extent of savings.

Finally, while a detailed interpretation of all network dynamics is beyond the scope of this work, we note that the cerebellum, despite its established role in motor adaptation [2,119–121,127–129], showed no eccentricity changes at the group level. This likely reflects a known property of dimension-reduction techniques (like PCA) to overweight variance

from structures with larger numbers of ROIs (i.e., the 400-parcel cortex versus the 32-parcel cerebellum) [74,76]. Thus, the stability of the cerebellum in our manifold space should not be interpreted as a lack of involvement, but rather as a consequence of our cortex-weighted analytical approach. Future studies employing targeted sub-network dimensionality reduction or focusing specifically on cerebellar dynamics will be required to fully characterize its contribution within this manifold framework. To this end, we did perform a targeted control analysis regressing out visual cortical signals from cerebellar time-series to ensure that our results were not driven by signal spillover from adjacent visual regions; our primary findings remained robust to this correction (S10 Fig).

In conclusion, our findings demonstrate that motor savings in humans is supported by the reinstatement of DMN FC patterns expressed during initial learning. This positions the DMN as a pivotal brain system in the retrieval and adaptive reuse of motor memories, extending its known roles in cognitive memory to the motor domain. Moreover, this re-expression of context-dependent neural states suggests a conserved neural mechanism across different memory modalities (e.g., motor learning, fear conditioning, episodic memory) for flexibly accessing and utilizing stored information.

## Materials and methods

### Participants

Forty right-handed individuals (27 females, aged 18−28 years; $M = 22.5$, SD = 4.51) initially participated in the MRI study. Right-handedness was assessed using the Edinburgh Handedness Questionnaire [126]. Of these 40 participants, eight individuals were removed from the final analysis ($N = 32$ included). Four participants were removed due to excessive head motion in the MRI scanner (motion greater than 2 mm or 2° rotation within a single scan) or missing volumes in a large portion of one of the task scans. An additional four participants were removed due to poor task compliance (i.e., >25% of trials not being completed within the maximal trial duration). Participants provided written informed consent before beginning the experimental protocol. The Queen's University Research Ethics Board approved the study (reference #: CNS-019-16) and it was conducted in coherence to the principles outlined in the Canadian Tri-Council Policy Statement on Ethical Conduct for Research Involving Humans and the principles of the Declaration of Helsinki (1964).

### Apparatus

Participants performed target-directed hand movements by applying force with their right index finger and thumb onto an MRI-compatible isometric force sensor (ATI Technologies; resolution 0.02 N; sampling rate 500 Hz) mounted on a custom-built platform positioned over their waist (see Fig 1A). Visual stimuli were rear-projected using an LCD projector (NEC LT265DLP projector, 1,024 × 768 resolution, 60 Hz refresh rate) onto a screen mounted behind the participant inside the scanner bore. Participants viewed the stimuli through a mirror fixed to the MRI head coil, preventing direct sight of their hand.

### Visuomotor rotation (VMR) task

We employed a well-characterized center-out VMR task [2,77] to probe sensorimotor adaptation and savings across two days. On each trial, participants controlled a cursor (20-pixel radius) displayed on the screen. To start a trial, a target circle (30-pixel radius) appeared at one of eight pseudo-randomized locations (0°, 45°, 90°, 135°, 180°, 225°, 270°, 315°) on an invisible ring (300-pixel radius) around the start position. The target filled in (white) after a 200 ms delay, cueing the participant to apply a brief, "shooting" force pulse to the sensor to propel the cursor through the target. The cursor traveled the 300-pixel distance over a fixed 750 ms duration with a bell-shaped velocity profile, regardless of the force magnitude (provided it exceeded a minimum threshold). Once the cursor reached the ring, it remained stationary for 1 s, providing endpoint error feedback. A "hit" was registered if the cursor overlapped with the target (target turned green). Each trial lasted 4 s, including movement time and feedback; any remaining time involved waiting for the next trial. Trials with

reaction times <100 ms or >2000 ms, or failure to initiate movement within the 2.6 s window after target presentation, were rare (<1% combined) and excluded from behavioral analyses but retained for continuous fMRI analysis.

## Procedure

Participants completed two identical fMRI testing sessions, scheduled exactly 24 hours apart. The core of each session was a continuous functional scan lasting 29 min and 52 s, during which the VMR task was performed. This task scan was structured sequentially into two main blocks: (1) Baseline Block: Consisted of 120 trials (structured as 15 consecutive sets of the 8 target directions) where the cursor feedback was veridical (i.e., no rotation applied); and (2) Learning Block: Consisted of 320 trials where a 45° CW rotation was applied to the cursor relative to the direction of applied force. Following this main task scan, participants underwent a subsequent functional scan (Washout block) lasting 8 min and 32 s. During this Washout block, subjects completed 120 trials wherein the VMR was removed (no rotation), and conditions reverted to the baseline state. Additionally, within each daily session, three 6-min resting-state fMRI scans were acquired: one prior to the main task scan, one immediately after the main task scan, and one following the washout scan. During these scans, participants were instructed to remain still with their eyes open, fixating on a central cross displayed on the screen.

This comprehensive two-day protocol was designed to permit investigation of adaptation, de-adaptation (washout), re-adaptation (Day 2), and individual differences therein across sessions. For the present study focusing on motor savings, we analyzed data from the task scans (Baseline, Learning, and Washout blocks) across both Day 1 and Day 2.

## MRI acquisition

Participants were scanned using a 3T Siemens TIM MAGNETOM Trio MRI scanner located at the Centre for Neuroscience Studies, Queen's University (Kingston, Ontario, Canada). For each participant on each day, we collected a T1-weighted ADNI MPRAGE anatomical (TR = 1,760 ms, TE = 2.98 ms, field of view = 192 mm × 240 mm × 256 mm, matrix size = 192 × 240 × 256, flip angle = 9°, 1 mm isotropic voxels). fMRI volumes were acquired using a 32-channel head coil and a T2*-weighted single-shot gradient-echo echo-planar imaging (EPI) acquisition sequence (TR = 2,000 ms, slice thickness = 4 mm, in-plane resolution = 3 mm × 3 mm, TE = 30 ms, field of view = 240 mm × 240 mm, matrix size = 80 × 80, flip angle = 90°), and acceleration factor (integrated parallel acquisition technologies, iPAT) = 2 with generalized auto-calibrating partially parallel acquisitions (GRAPPA) reconstruction. Each volume comprised 34 contiguous (no gap) oblique slices acquired at a 30° caudal tilt with respect to the plane of the anterior and posterior commissure (AC-PC), providing whole-brain coverage. For the baseline and learning scan, we collected a single, continuous scan of 896 imaging volumes. For the washout scan, we collected a single scan of 256 imaging volumes. Both of these scans included an additional 8 imaging volumes at both the beginning and the end of the experimental run.

## fMRI preprocessing

Preprocessing of anatomical and fMRI data was performed using fMRIPrep 20.1.1 [130,131] (RRID:SCR_016216) which is based on Nipype 1.5.0 [132,133] (RRID:SCR_002502). Many internal operations of fMRIPrep use Nilearn 0.6.2 [134] (RRID:SCR_001362), mostly within the functional processing workflow. For more details of the pipeline, see the section corresponding to workflows in fMRIPrep's documentation. Below we provide a condensed description of the preprocessing steps (See [61,74] for previous descriptions of this same imaging approach).

T1w images were corrected for intensity non-uniformity (INU) with N4BiasFieldCorrection [135], distributed with ANTs 2.2.0 [136] (RRID:SCR_004757). The T1w-reference was then skull-stripped with a Nipype implementation of the ants-BrainExtraction.sh workflow (from ANTs), using OASIS30ANTs as target template. Brain tissue segmentation of cerebrospinal fluid (CSF), white matter (WM), and gray matter (GM) was performed on the brain-extracted T1w using fast (FSL 5.0.9, RRID:SCR_002823) [137]. A T1w-reference map was computed after registration of the T1w images (after

INU-correction) using mri_robust_template (FreeSurfer 6.0.1) [138]. Brain surfaces were reconstructed using recon-all (FreeSurfer 6.0.1, RRID:SCR_001847) [139], and the brain mask estimated previously was refined with a custom variation of the method to reconcile ANTs-derived and FreeSurfer-derived segmentations of the cortical GM of Mindboggle (RRID:SCR_002438) [140]. Volume-based spatial normalization to standard space (MNI152NLin6Asym) was performed through nonlinear registration with antsRegistration (ANTs 2.2.0), using brain-extracted versions of both T1w reference and the T1w template.

For each BOLD run, the following preprocessing was performed. First, a reference volume and its skull-stripped version were generated using a custom methodology of fMRIPrep. The BOLD reference was then co-registered to the T1w reference using bbregister (FreeSurfer) which implements boundary-based registration [141]. Co-registration was configured with nine degrees of freedom to account for distortions remaining in the BOLD reference. Head-motion parameters with respect to the BOLD reference (transformation matrices, and six corresponding rotation and translation parameters) are estimated before any spatiotemporal filtering using MCFLIRT (FSL 5.0.9, [142]). BOLD runs were slice-time corrected using 3dTshift from AFNI 20160207 ([143], RRID:SCR_005927). The BOLD time-series, were resampled to surfaces on the following spaces: fsaverage. The BOLD time-series (including slice-timing correction when applied) were resampled onto their original, native space by applying a single, composite transform to correct for head-motion and susceptibility distortions. These resampled BOLD time-series will be referred to as preprocessed BOLD in original space, or just preprocessed BOLD. The BOLD time-series were resampled into several standard spaces, correspondingly generating the following spatially-normalized, preprocessed BOLD runs: MNI152NLin6Asym, MNI152NLin2009cAsym. First, a reference volume and its skull-stripped version were generated using a custom methodology of fMRIPrep. Automatic removal of motion artifacts using independent component analysis (ICA-AROMA, [144]) was performed on the preprocessed BOLD on MNI space time-series after removal of non-steady state volumes and spatial smoothing with an isotropic, Gaussian kernel of 6 mm FWHM (full-width half-maximum). Corresponding "non-aggressively" denoised runs were produced after such smoothing. Additionally, the "aggressive" noise-regressors were collected and placed in the corresponding confounds file. Several confounding time-series were calculated based on the preprocessed BOLD: framewise displacement (FD), DVARS, and three region-wise global signals. FD and DVARS are calculated for each functional run, both using their implementations in Nipype (following the definitions by [145]). The three global signals are extracted within the CSF, the WM, and the whole-brain masks. Additionally, a set of physiological regressors were extracted to allow for component-based noise correction (CompCor, [146]). Principal components are estimated after high-pass filtering the preprocessed BOLD time-series (using a discrete cosine filter with 128s cut-off) for the two CompCor variants: temporal (tCompCor) and anatomical (aCompCor). tCompCor components are then calculated from the top 5% variable voxels within a mask covering the subcortical regions. This subcortical mask is obtained by heavily eroding the brain mask, which ensures it does not include cortical GM regions. For aCompCor, components are calculated within the intersection of the aforementioned mask and the union of CSF and WM masks calculated in T1w space, after their projection to the native space of each functional run (using the inverse BOLD-to-T1w transfor-mation). Components are also calculated separately within the WM and CSF masks. For each CompCor decomposition, the k components with the largest singular values are retained, such that the retained components' time series are sufficient to explain 50 percent of variance across the nuisance mask (CSF, WM, combined, or temporal). The remaining components are dropped from consideration. The head-motion estimates calculated in the correction step were also placed within the corresponding confounds file. The confound time series derived from head motion estimates and global signals were expanded with the inclusion of temporal derivatives and quadratic terms for each [147]. Frames that exceeded a threshold of 0.5 mm FD or 1.5 standardized DVARS were annotated as motion outliers. All resamplings can be performed with a single interpolation step by composing all the pertinent transformations (i.e., head-motion transform matrices, susceptibility distortion correction when available, and co-registrations to anatomical and output spaces). Gridded (volumetric) resamplings were performed using antsApplyTransforms (ANTs), configured with Lanczos interpolation to minimize the smoothing effects of other kernels [148]. Non-gridded (surface) resamplings were performed using mri_vol2surf (FreeSurfer).

**Regional time series extraction.** For each participant and scan, the average BOLD time series were computed from the grayordinate time series for 464 regions defined by combining: (1) the Schaefer 400 cortical parcellation [79], (2) the Tian scale II 3T subcortical atlas (32 regions including hippocampus, amygdala, basal ganglia, thalamus) [80], and (3) the Nettekoven medium-granularity cerebellar atlas (32 regions covering motor, action, demand, and sociolinguistic functions) [81]. Time series were denoised using the confound regressors described above (motion, aCompCor, etc.) plus discrete cosine regressors (128s cut-off for high-pass filtering) produced by fMRIprep, and then low-pass filtered (Butterworth filter, 100s cut-off) using Nilearn. Finally, all region time series were z-scored. See our previous work [61,74] for similar approaches.

**Cerebellar control analysis (visual nuisance regression).** We addressed the potential for signal contamination of cerebellar regions by adjacent ventral occipital cortex—a concern highlighted by their spatial proximity and by similar loadings along the second principal component of the baseline manifold (Fig 2A). To rule out signal leakage, we performed an adjacent-tissue nuisance regression control in the spirit of established approaches for mitigating cerebellar–occipital partial-volume and shared-variance artifacts [89,149]. Specifically, we defined ventral visual nuisance signals using the Schaefer2018 cortical parcellation (400 parcels, 7 networks) by selecting parcels assigned to the Visual network whose centroid z-coordinate fell below a ventral "floor" threshold ($z < -4$; centroid table provided with the atlas).

From these ventral visual parcels, we extracted parcel-wise BOLD time series, applied PCA, and retained the top three components (PC1–PC3), which cumulatively explained $51 \pm 3\%$ of variance across subjects. These components were then z-scored and regressed (with an intercept) from the time series of all cerebellar parcels (32 regions) prior to estimating connectivity and recomputing manifold measures. All downstream analyses (including inferential testing) were then repeated on these "cleaned" cerebellar signals. This control analysis yielded results highly consistent with the original analysis (spatial correlation of F-statistic maps: $r = 0.95$, $p < 0.001$; S10 Fig), confirming that our findings are robust to potential visual signal spillover.

## Neuroimaging data analysis

**Covariance estimation and centering.** For every participant, denoised region time series from the functional task scans on Day 1 and Day 2 were each spliced into five equal-length task epochs (96 imaging volumes/48 trials each or 6 eight-trial bins), after discarding the first 6 imaging volumes. The epochs were: Baseline, Early Learning (first 48 trials of rotation), Late Learning (last 48 trials of rotation), Early Washout (first 48 trials of washout), and Late Washout (last 48 trials of washout). FC matrices for each epoch were estimated by computing the region-wise covariance matrix using the Ledoit–Wolf estimator [150]. Note that our use of equal-length epochs for the six phases ensured that no biases in covariance estimation were introduced due to differences in time series length.

Next, we centered the connectivity matrices using the Riemannian manifold approach [61,74–76,84,85,151] to remove static, subject-specific variations. First, a grand mean covariance matrix, $\overline{S}_{gm}$, was computed via the geometric mean across all participants and all 10 epochs (5 epochs × 2 days). Then, for each participant $i$, the geometric mean across their 10 epochs, $\overline{S}_i$, was computed. Each epoch-specific covariance matrix $S_{ij}$ (for participant $i$, epoch $j$) was projected to the tangent space at $\overline{S}_i$ as $T_{ij} = \overline{S}_i^{1/2} log(\overline{S}_i^{-1/2} S_{ij} \overline{S}_i^{-1/2}) \overline{S}_i^{1/2}$. Each $T_{ij}$ was transported to the grand mean $\overline{S}_{gm}$ via $T_{ij}^c = GT_{ij}G^T$, where $G = \overline{S}_{gm}^{1/2} \overline{S}_i^{-1/2}$. Finally, each centered tangent vector $T_{ij}^c$ was projected back to obtain the centered covariance matrix $S_{ij}^c = \overline{S}_{gm}^{1/2} exp(\overline{S}_{gm}^{-1/2} T_{ij}^c \overline{S}_{gm}^{-1/2}) \overline{S}_{gm}^{1/2}$. For the benefits of this centering approach, see Fig 1D, and for an additional overview, see [74,85].

**Manifold construction and eccentricity calculation.** Following centering, connectivity manifolds were derived [61,64,71,88]. For each centered covariance matrix, we applied row-wise thresholding (top 10% of connections), computed the cosine similarity between rows to create an affinity matrix, and performed PCA on the affinity matrix. To provide a common reference space, we constructed a template Baseline manifold using the geometric mean of all participant-averaged Baseline matrices (average of Day 1 and Day 2 Baseline). All individual manifolds (32

participants × 10 epochs = 320 total) were aligned to this template using Procrustes alignment. All subsequent analyses used the top three PCs, which cumulatively explained ~46% of the template manifold variance (Fig 2B). Eccentricity was calculated for each region in each aligned manifold as its Euclidean distance from the manifold centroid (origin: [0,0,0]) in the 3D space defined by PCs 1–3 [61,73,74,90,91,152]. Note that the results of our analyses were not qualitatively different if we also included PCs 4–10 in our analyses (see S1 Fig).

**Representational similarity analysis.** To test the specific hypothesis that motor savings involves the reinstatement of the initial learning state, we used Representational Similarity Analysis [92,153,154]. Our approach was targeted and hypothesis-driven, focusing specifically on the relationship between the initial learning epoch and all subsequent task epochs. For each participant, we therefore calculated the Pearson spatial correlation between the vector of regional eccentricity values during Day 1 Early Learning and the vector of values during each of the subsequent seven learning-related epochs (Day 1 Late Learning through Day 2 Late Washout). This yielded a similarity profile showing how closely each subsequent epoch's manifold structure resembled the initial learning state. These comparisons were performed separately within each of the four functional ensembles identified via k-means clustering (see Fig 3D). Pearson correlation was used as the primary similarity metric because it is the standard measure in RSA and provides a sensitive and interpretable estimate of linear similarity between high-dimensional patterns. To confirm that our key reinstatement result was not dependent on distributional assumptions or outliers, we additionally repeated the RSA using a rank-based Spearman correlation as a control analysis (S11 Fig). This control analysis yielded the same qualitative pattern and preserved the selective DMN-centric reinstatement effect (Spearman-based rmANOVA: $F(6, 186) = 2.21$, $p = 0.035$; planned contrasts: all $ps < 0.05$, see S11 Fig). Statistical significance was assessed using repeated measures ANOVAs and one-sided paired $t$-tests comparing the Day 1 Early Learning versus Day 2 Early Relearning similarity against the similarity between Day 1 Early Learning and other epochs.

**Learner subgroup analysis.** To assess learner heterogeneity, participants were median-split into "fast" and "slow" learners based on the Learning Score (no outliers excluded). The RSA procedure described above was then repeated separately within each subgroup, comparing the spatial pattern of regional eccentricity during Day 1 Early Learning to all subsequent learning and washout epochs for each functional ensemble.

**Eccentricity analyses.** To identify adaptation-related changes, we performed region-wise 2 (Day: 1, 2) × 5 (Epoch: Baseline, Early Learning, Late Learning, Early Washout, Late Washout) repeated measures ANOVAs on eccentricity values. Results were corrected for multiple comparisons using FDR ($q < 0.05$). Post-hoc paired $t$-tests (FDR corrected) were used to examine specific epoch contrasts averaged across days (e.g., Early > Baseline, Late > Early, etc., reflecting the main effect of Epoch) as shown in Fig 3E. See our previous work [61,74–76] for a description of similar approaches.

**Seed connectivity analyses.** To interpret eccentricity changes, we performed seed-based connectivity analyses using representative regions across the four functional ensembles (Fig 3D). Seeds included left PCC and left MPFC (DMN-centric), left M1 (Sensorimotor-centric), and left lateral visual cortex (Visual-centric). For each seed, we computed average connectivity maps for Baseline (avg. Day 1 and 2), Early (avg. Day 1 Early Learning and Day 2 Early Relearning), and Late (avg. Day 1 Late Learning and Day 2 Late Relearning) epochs. We performed paired t-tests between these average connectivity maps for each epoch (e.g., Early > Baseline, Late > Early) across participants to generate unthresholded statistical maps visualizing multivariate connectivity changes. Network-level changes were summarized using spider plots based on the Yeo 7-network parcellation [89].

## Behavioral data analysis

**Data preprocessing.** Visuomotor errors were measured as the angular difference between the target location and the final cursor endpoint on each trial. Trials with reaction times <100 ms or >2000 ms, or failure to initiate movement within the 2.6 s window after target presentation, were excluded from behavioral analyses (<1% combined). Initial error for Day 1 Learning and Day 2 Relearning was defined as the median angular error over the first 16 trials (two full target cycles) of

the respective Learning blocks. Behavioral savings was calculated as the difference between Day 1 initial error and Day 2 initial error [9,94,consistent with prior work 99].

**Learning score derivation (fPCA).** To derive a continuous measure capturing individual differences in overall learning proficiency across the entire two-day experiment, we applied fPCA to the binned participants' angular error trajectories. This approach allows us to identify the dominant modes of variation in how subjects' performance evolved over the course of learning, relearning, and washout on both days.

First, for each participant, their complete behavioral data (trial-by-trial angular error) across all phases of Day 1 and Day 2 were averaged into eight-trial bins to provide a smoother representation of their learning curve. We then represented each participant's binned learning curve as a continuous function using a cubic B-spline basis with 17 basis functions (order 4; with knots placed at the center of each bin and smoothing enforced by a second-derivative penalty). This step effectively converted each discrete learning curve into a smooth functional data object.

Next, we performed fPCA on this set of individual functional learning curves [155]. Just as standard PCA estimates a low-dimensional subspace capturing variability in multivariate data, fPCA approximates a set of functions (our learning curves) using a smaller set of orthogonal basis functions (functional principal components, fPCs). This approach is particularly suitable when observations represent continuous processes, as the basis expansion used in fPCA can inherently encode features like smoothness.

The fPCA decomposes the variability across all subjects' learning curves into a set of fPCs, each explaining a decreasing amount of the total variance. In our analysis, the first fPC captured the dominant pattern of inter-subject variability in overall learning and adaptation performance, explaining a substantial majority of the variance (~63%). Because the sign of principal components is arbitrary, we multiplied the FPC1 loadings by −1 so that higher Learning Scores correspond to lower angular error (better performance) across the experiment. The score of each participant onto this first fPC was therefore taken as their "Learning Score," providing a single, continuous measure of overall learning proficiency throughout the two-day experiment (Fig 6D). All spline smoothing and fPCA procedures were performed using the *scikit-fda* [156] Python package.

**Behavioral correlation analysis.** We investigated the relationship between manifold structure and individual differences in task performance. First, we correlated the Learning Score with the change in manifold eccentricity (Early > Baseline contrast) for each brain region, separately for Day 1 and Day 2. Second, these region-wise correlations were evaluated at the network level by calculating the average eccentricity change within each of the Yeo 7 networks [89] and correlating this with the Learning Score. Statistical significance for network-level correlations was assessed using 1,000 iterations of the Váša spin-test permutation procedure [103,104] and corrected using FDR ($q < 0.05$). Third, to interpret significant network-level correlations, we performed exploratory whole-brain seed-to-network connectivity analyses, correlating the Learning Score with the change in connectivity (Early > Baseline) between the significant network (e.g., DMN) and all other regions/networks.

**Recall Ratio analysis.** Following [102], we computed a RR to quantify the extent to which early Day 2 performance approached the learned state at the end of Day 1 adaptation (i.e., early recall normalized by late discovery performance). Because our behavioral measure is signed angular error and the imposed rotation was 45°, we first converted error to an adaptation-performance proxy $A = 45° - |error|$, such that larger values indicate closer approach to the fully adapted state. RR was then defined as: $RR = A_{D2\ Early} / A_{D1\ Late}$. RR ≈ 1 indicates near-complete recall of the previously learned state, RR < 1 indicates partial recall, and RR > 1 indicates expression of a larger correction upon re-exposure. To ensure interpretability when late Day 1 adaptation was negligible, we applied a stability threshold of 5° when computing the ratio; no subjects were excluded from any other analyses. We tested associations between RR and (i) behavioral savings and (ii) Learning Score using Pearson correlation. To examine neural correlates of RR using the same approach as Fig 7, we correlated RR with baseline-corrected eccentricity values during Day 1 Late Learning within each canonical functional network and assessed significance using the same permutation-testing framework used for the Learning Score analysis.

## Software

Imaging data were preprocessed using fmriPrep [130], which is open source and freely available. All analyses were performed using Python 3.11.6 and involved the following open-source Python packages. FC estimation and centering were performed with Nilearn 0.10.2 [134] and PyRiemann 0.6 [157], respectively. All steps to generate and align connectivity manifolds were carried out using Brainspace 0.1.1 [71]. Graph theoretical measures were computed using the Brain Connectivity Toolbox in Python (bctpy 0.6.1; https://github.com/aestrivex/bctpy/wiki), and spin permutation testing procedures were implemented in neuromaps 0.0.5 [103]. All statistical analyses were performed with Pingouin 0.5.5 [158] and SciPy 1.11.4 [159]. For unsupervised learning analyses, UMAP was implemented with Umap-learn 0.5.5 [86], and k-means clustering was performed with Scikit-learn 1.5.1 [160]. Surface visualizations were generated using Surfplot 0.2.0 [161]. Representational similarity analysis was conducted using rsatoolbox 0.1.5 [153]. General data processing and visualization were performed using NumPy 1.26.2 [162], Pandas 2.2.1 [163], Nibabel 5.2.0 [164], Matplotlib 3.10.1 [165], Seaborn 0.13.2 [166], and Cmasher 1.9.2 [167]. The analysis code is archived on Zenodo (DOI: https://doi.org/10.5281/zenodo.18612771), also available at https://github.com/alirzar/motorsaving.

## Supporting information

**S1 Fig. Comparison of learning-related changes in manifold eccentricity using different numbers of principal components (PCs). (A–C)** Brain maps showing the main effect of Task Epoch on manifold eccentricity, calculated using the top 3 PCs (A), top 4 PCs (B), and top 5 PCs (C), respectively. The results show a high degree of correspondence in the significant effects across the different numbers of PCs. **(D)** Effect of including lower variance-explained PCs on the main effect brain maps. The plot shows the spatial correlation between the main effect map from our primary analysis (using 3 PCs) and maps generated using a progressively larger number of PCs (from 4 to 10). Note the high degree of spatial correlation, even as additional PCs are added to the data.
(TIF)

**S2 Fig. Functional connectivity properties that underlie manifold eccentricity.** For three different graph theoretical measures—node strength **(A)**, participation coefficient **(B)**, and within-module degree z-score **(C)**—the top panels show the spatial map of each measure derived from the group-average Baseline connectivity matrix. The bottom panels show the corresponding scatterplot relating each measure to regional eccentricity. We found that Baseline eccentricity was positively correlated with node strength ($r = 0.78$, $p < 0.001$), positively correlated with within-module degree z-score ($r = 0.53$, $p < 0.001$), and negatively correlated with the participation coefficient, a measure of cross-network integration ($r = -0.59$, $p < 0.001$). Together, these results support the idea that areas with higher eccentricity generally have stronger functional coupling with other parts of the same functional network (i.e., higher segregation) whereas areas with lower eccentricity generally have stronger connectivity across different networks (i.e., higher integration).
(TIF)

**S3 Fig. Task Epoch-related connectivity changes do not reflect univariate changes in brain activity.** The plots show the mean BOLD activation (z-scored time series data, averaged across participants) for each task epoch within the regions that exhibited a significant main effect of Task Epoch in our primary connectivity analysis. The line overlay shows the group average across these regions, color-coded by day (Day 1: cyan; Day 2: orange). A two-way repeated measures ANOVA (Day x Task Epoch) on this activation data revealed no significant effects in any region after FDR correction ($q < 0.05$). This null result indicates that the manifold eccentricity changes described in the main text are not driven by simple changes in mean BOLD amplitude. The plots show mean BOLD activation (within-epoch mean of z-scored time series) for each task epoch within regions that exhibited a significant main effect of Task Epoch in our primary

connectivity analysis. Underlying numerical data are provided in S1 Data and archived on Zenodo (https://doi.org/10.5281/zenodo.18613054).
(TIF)

**S4 Fig. Eccentricity modulation across task epochs (unthresholded maps).** This figure shows the unthresholded statistical maps from the pairwise *t*-tests presented in Fig 3E. The maps compare eccentricity between task epochs, averaged across Day 1 and Day 2. Red indicates expansion (increased eccentricity), while blue indicates contraction (decreased eccentricity).
(TIF)

**S5 Fig. Connectivity changes underlying the Task Epoch effect for right hemisphere seed regions.** This figure illustrates changes in whole-brain functional connectivity for representative seed regions (indicated in yellow and by arrows) selected based on the significant main effect of Task Epoch from the ANOVA (see Fig 3A). Each row corresponds to a specific seed region. **(A)** Brain maps display the results of paired t-tests contrasting connectivity patterns between different task epochs (averaged across Day 1 and Day 2). For a given contrast (e.g., Early > Baseline) red denotes increased connectivity in the first epoch relative to the second, while blue denotes decreased connectivity. Specific contrasts are shown above the maps. **(B)** Scatter plots show the manifold eccentricity trajectory for the corresponding seed region across all 10 task epochs. Individual points represent participant means per epoch, color-coded by Day (Day 1: cyan, Day 2: orange). The solid line overlay indicates the group mean, illustrating the consistency of eccentricity changes across both days. **(C)** Spider plots summarize the network-level profile of the connectivity changes shown in panel (A), aggregated according to the Yeo 7-network parcellation (77). The radial axis represents the t-statistic for the change in connectivity between the seed region and each target network for the specified contrast (e.g., Early > Baseline). Points outside the black circle ($t = 0$; no change) indicate an increase in connectivity for that contrast, while points inside the circle indicate a decrease. Underlying numerical data are provided in S1 Data and archived on Zenodo (https://doi.org/10.5281/zenodo.18613054).
(TIF)

**S6 Fig. Connectivity changes that underlie the Task Epoch effect for Washout (WO) epochs.** This figure illustrates changes in whole-brain functional connectivity for representative seed regions (indicated in yellow and by arrows) selected based on the significant main effect of Task Epoch from the ANOVA (same regions from Fig 5). Each row corresponds to a specific seed region. **(A)** Brain maps display the results of paired t-tests contrasting connectivity patterns between different washout task epochs (averaged across Day 1 and Day 2). For a given contrast (e.g., WO-early > Baseline), red denotes increases in connectivity in the first epoch relative to the second, while blue denotes decreased connectivity. Specific contrasts are shown above the maps. Note that the connectivity changes observed across the seed regions resemble those seen during learning (Fig 5), though they are comparatively more subtle. **(B)** Scatter plots show the manifold eccentricity trajectory for the corresponding seed region across all 10 task epochs. Individual points represent participant means per epoch, color-coded by Day (Day 1: cyan, Day 2: orange). The solid line overlay indicates the group mean, illustrating the consistency of eccentricity changes across both days. **(C)** Spider plots summarize the network-level profile of the connectivity changes shown in panel A, aggregated according to the Yeo 7-network parcellation (77). The radial axis represents the t-statistic for the change in connectivity between the seed region and each target network for the specified contrast (e.g., WO-early > Baseline). Points outside the black circle ($t = 0$; no change) indicate an increase in connectivity for that contrast, while points inside the circle indicate a decrease. Underlying numerical data are provided in S1 Data and archived on Zenodo (https://doi.org/10.5281/zenodo.18613054).
(TIF)

**S7 Fig. Network eccentricity and connectivity changes during Washout correlating with individual learning performance. (A)** Whole-brain map displaying correlations between participants' Learning Scores and regional changes in manifold eccentricity during initial washout (WO1 Early > Day 1 Baseline). **(B)** Spatial permutation testing ("spin-test") identifies functional networks whose Day 1 eccentricity changes (from A) significantly correlate with Learning Scores. Each data point shows the actual correlation for one of 7 Yeo and colleagues [1] networks. Boxplots depict the null correlation distribution (1000 iterations) for each network [2, 3]. Boxplot elements: center line = median; box edges = 25th/75th quartiles; whiskers = min-max of null distribution. Dashed blue line indicates $r = 0$. The data points in red denote significant network-specific correlations (FDR corrected, $q < 0.05$). **(C)** Scatterplot illustrating the correlation from panel (A) for an example brain region (indicated in yellow) with participants' Learning Scores on Day 1. **(D)** Whole-brain correlation map as in (A), but for eccentricity changes during early Washout on Day 2 (WO2 Early > Day 2 Baseline). **(E)** Network-level spin-test results as in (B), but for the Day 2 correlations shown in (D). **(F)** Scatterplot illustrating the correlation from panel (E) for an example brain region on Day 2. **(G)** Brain maps show how the strength of connectivity change for the DMN, Control, and SalVentAttn networks correlates with the Learning Score. Red indicates that increase in connectivity with a given region is associated with better performance (higher Learning Score). Blue indicates that increased connectivity is associated with worse performance (lower Learning Score). Underlying numerical data are provided in S1 Data and archived on Zenodo (https://doi.org/10.5281/zenodo.18613054).
(TIF)

**S8 Fig. Network eccentricity and connectivity changes correlating with individual learning performance (for non-DMN networks). (A)** Whole-brain map displaying correlations between participants' Learning Scores and regional changes in manifold eccentricity during initial learning (Day 1 Early Learning > Day 1 Baseline). **(B)** Spatial permutation testing ("spin-test") identifies functional networks whose Day 1 eccentricity changes (from A) significantly correlate with Learning Scores. Each point shows the actual correlation for one of the 7 Yeo and colleagues [1] networks. Boxplots depict the null correlation distribution (1,000 iterations) for each network [2, 3]. Boxplot elements: center line = median; box edges = 25th/75th quartiles; whiskers = min-max of null. Dashed blue line indicates $r = 0$. Data points in red denote significant network-specific correlations (FDR corrected, $q < 0.05$). **(C)** Brain map illustrating how Day 1 changes in functional connectivity between the DorsAttn network (top) and Somatomotor network (bottom) and all other brain regions correlate with the Learning Score. Red indicates that increased inter-network connectivity with the DorsAttn network (top) or Somatomotor network (bottom) is related to higher Learning Scores (better performance); blue indicates that increased connectivity is associated with lower Learning Scores. **(D)** Whole-brain correlation map as in (A), but for eccentricity changes during early relearning on Day 2 (Day 2 Early Relearning > Day 2 Baseline). **(E)** Network-level spin-test results as in (B), but for the Day 2 correlations shown in (D). **(F)** Brain map as in (C), but showing how Day 2 changes in the connectivity of the Frontoparietal Control network correlate with Learning Score. Underlying numerical data are provided in S1 Data and archived on Zenodo (https://doi.org/10.5281/zenodo.18613054).
(TIF)

**S9 Fig. Recall Ratio analysis. (A)** Distribution of Recall Ratio (RR) values across participants, defined as ($RR = Adaptation_{D2\ Early} / Adaptation_{D1\ Late}$). **(B)** Scatterplots relating RR to behavioral savings (left) and Learning Score (right), showing significant associations with each metric (RR–Saving: $r = 0.48$, $p < 0.01$; RR–Learning Score: $r = 0.76$, $p < 0.001$). **(C)** Network-level correlations between RR and baseline-corrected manifold eccentricity during Day 1 Late Learning. Correlations were computed separately within each canonical functional network and assessed using the same permutation-testing framework used in Fig 7. While higher Recall Ratios were generally associated with greater manifold contraction (negative correlations) across most networks, no single network exhibited a statistically significant selective effect after permutation testing. Underlying numerical data are provided in S1 Data and archived on Zenodo (https://doi.org/10.5281/zenodo.18613054).
(TIF)

**S10 Fig. Cerebellar control analysis for potential visual signal contamination. (A)** Schematic of the nuisance-regression pipeline: ventral visual cortex time series were treated as a nuisance source, PCA was used to extract the top three visual components, and these components were regressed from cerebellar time series to yield a cleaned cerebellar signal. **(B)** Spatial loadings for the top three visual PCs for the nuisance-regressed data. **(C)** Whole-brain manifold structure in low-dimensional PC space for the nuisance-regressed data. **(D)** Manifold organization/eccentricity maps following visual-component regression. **(E)** Robustness of inferential results: parcel-wise F-statistics from the original ANOVA strongly correlate with *F*-statistics from the visual-corrected analysis ($r = 0.95$, $p < 0.001$), indicating that manifold changes are nearly identical across analysis pipelines. The dashed line is the unity line.
(TIF)

**S11 Fig. RSA results are robust to correlation metric (Spearman control).** To confirm that the reinstatement effects were not driven by outliers or distributional assumptions inherent to Pearson correlation, we repeated the representational similarity analysis (RSA) using rank-based Spearman correlations. **(A)** The full-sample RSA results shown in Fig 4 using Spearman correlation, plotting the similarity between the Day 1 Early Learning eccentricity pattern and each subsequent learning/washout epoch separately for each functional ensemble (DMN-centric, Visual-centric, Sensorimotor-centric, SalVentAttn-centric). **(B)** The learner heterogeneity RSA results shown in Fig 8 using Spearman correlation, plotted separately for fast and slow learners (median split on Learning Score). In all plots, the black line shows the across-subject mean and individual points show single subjects, color-coded by day (Day 1: cyan; Day 2: orange). Asterisks denote significant effects from one-tailed paired-samples t-tests comparing the similarity between Day 1 Early Learning and Day 2 Early Relearning versus other epochs (* $p < 0.05$, ** $p < 0.01$). Across both the full sample and subgroup analyses, the key finding—a selective DMN-centric reinstatement profile with elevated similarity at Day 2 Early Relearning—remained qualitatively and statistically consistent when using Spearman correlation. Underlying numerical data are provided in S1 Data and archived on Zenodo (https://doi.org/10.5281/zenodo.18613054).
(TIF)

**S1 Data. Underlying numerical data for figures (Excel workbook).** Archived on Zenodo (https://doi.org/10.5281/zenodo.18613054).
(XLSX)

## Acknowledgments

The authors would like to thank Martin York and Sean Hickman for technical assistance, and Don O'Brien for assistance with data collection.

## Author contributions

**Conceptualization:** Ali Rezaei, Corson N. Areshenkoff, Jason P. Gallivan.

**Data curation:** Ali Rezaei, Corson N. Areshenkoff, Jason P. Gallivan.

**Formal analysis:** Ali Rezaei.

**Funding acquisition:** Jason P. Gallivan.

**Investigation:** Ali Rezaei, Corson N. Areshenkoff, Jason P. Gallivan.

**Methodology:** Ali Rezaei, Corson N. Areshenkoff.

**Project administration:** Jason P. Gallivan.

**Resources:** Ali Rezaei, Corson N. Areshenkoff, Daniel J. Gale, Jason P. Gallivan.

**Software:** Ali Rezaei, Daniel J. Gale.

**Supervision:** Jason P. Gallivan.

**Validation:** Ali Rezaei, Jason P. Gallivan.

**Visualization:** Ali Rezaei.

**Writing – original draft:** Ali Rezaei, Jason P. Gallivan.

**Writing – review & editing:** Ali Rezaei, Corson N. Areshenkoff, Emily R. Oby, Jonathan Smallwood, J. Randall Flanagan, Jeffrey D. Wammes, Jason P. Gallivan.

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
