## [Editor Report · Decision Letter 0]

17 Nov 2025

Dear Jason,

Thank you for submitting your manuscript entitled "Reinstatement of default mode network manifold structure underlies motor memory retrieval" for consideration as a Research Article by PLOS Biology.

Your manuscript has now been evaluated by the PLOS Biology editorial staff as well as by an academic editor with relevant expertise and I am writing to let you know that we would like to send your submission out for external peer review.

Once your full submission is complete, your paper will undergo a series of checks in preparation for peer review. After your manuscript has passed the checks it will be sent out for review. To provide the metadata for your submission, please Login to Editorial Manager (https://www.editorialmanager.com/pbiology) within two working days, i.e. by Nov 19 2025 11:59PM.

Kind regards,

Christian

Christian Schnell, PhD

Senior Editor

PLOS Biology

cschnell@plos.org

---

## [Editor Report · Decision Letter 1]

2 Dec 2025

Dear Jason,

Thank you for your patience while we discussed the revision plan of your manuscript "Reinstatement of default mode network manifold structure underlies motor memory retrieval", which had previously been reviewed at another journal, with the academic editor.

We would like to invite you to revise the work to thoroughly address the reviewers' reports, along the lines that you outlined in your proposal.

Given the extent of revision needed, we cannot make a decision about publication until we have seen the revised manuscript and your response to the reviewers' comments. Your revised manuscript is likely to be sent for further evaluation by all or a subset of the reviewers.

**IMPORTANT - SUBMITTING YOUR REVISION**

*Re-submission Checklist*

*Published Peer Review*

*PLOS Data Policy*

*Blot and Gel Data Policy*

*Prior review process*

Could you please forward me the decision letters from the prior journal, including handling editor and the manuscript ID, so we can contact the journal to validate the reviewer reports and share the reviewer identities with us?

Sincerely,

Christian

Christian Schnell, PhD

Senior Editor

PLOS Biology

cschnell@plos.org

---

## [Decision Letter · Decision Letter 2]

6 Feb 2026

Dear Jason,

Thank you for your patience while we considered your revised manuscript "Reinstatement of default mode network manifold structure underlies motor memory retrieval", submitted after review at a another journal under our portable peer review policy, for publication as a Research Article at PLOS Biology. This revised version of your manuscript has been evaluated by the PLOS Biology editors, the Academic Editor and two reviewers.

Based on the reviews and on our Academic Editor's assessment of your revision, we are likely to accept this manuscript for publication, provided you satisfactorily address the following data and other policy-related requests:

* We would like to suggest a different title to improve its accessibility for our broad audience:

The retrieval of previously learned motor memories is facilitated by the reinstatement of default mode network manifold structures

* Please include the approval/license number of the ethical approval for the experiments.

* We also want to encourage you to opt into transparent peer review, so readers are aware that your manuscript was transferred to us after peer review at another journal.

* DATA POLICY:

Regardless of the method selected, please ensure that you provide the individual numerical values that underlie the summary data displayed in the following figure panels as they are essential for readers to assess your analysis and to reproduce it: 3D, 4 (right panels), 5B, 7BF, 8AB (right panels), S3, S5B, S6B, S7BE, S8BE, S9AC and S11AB.

* CODE POLICY

Per journal policy, if you have generated any custom code during the course of this investigation, please make it available without restrictions. Please ensure that the code is sufficiently well documented and reusable, and that your Data Statement in the Editorial Manager submission system accurately describes where your code can be found. More information on our Code Policy, what and how to share can be found here: https://journals.plos.org/plosbiology/s/code-availability

As you address these items, please take this last chance to review your reference list to ensure that it is complete and correct (for example, references 106 and 108 are the same). If you have cited papers that have been retracted, please include the rationale for doing so in the manuscript text, or remove these references and replace them with relevant current references. Any changes to the reference list should be mentioned in the cover letter that accompanies your revised manuscript.

We expect to receive your revised manuscript within two weeks.

*Published Peer Review History*

*Press*

Sincerely,

Christian

Christian Schnell, PhD,

Senior Editor

cschnell@plos.org

PLOS Biology

Reviewer remarks:

Reviewer #1 (Jonathan Tsay): Thank you for your thoughtful revisions! Appreciate all the work put in to address my comments and concerns. I think the manuscript is a wonderful contribution to the field.

dD>

Reviewer #2: The authors have done an excellent and extensive job responding to both me and Reviewer 1. They have particularly done a good job with additional control analyses and with re-framing the study impact and motivation. This is a nice, rigorous study. I commend the authors on their excellent work.

---

## [Editor Report · Decision Letter 3]

18 Feb 2026

Dear Dr Gallivan,

My name is Luke Smith - I am an editor at PLOS Biology, and am keeping an eye on your manuscript "The retrieval of previously learned motor memories is facilitated by the reinstatement of default mode network manifold structures", while my colleague, Christian Schnell is out of the office this week. Thank you for addressing our last editorial requests in this most recent revision. On behalf of my colleagues, and the Academic Editor, Matthew F. S. Rushworth, I am pleased to say that we are satisfied by the changes made and can in principle accept your manuscript for publication, provided you address any remaining formatting and reporting issues. These will be detailed in an email you should receive within 2-3 business days from our colleagues in the journal operations team; no action is required from you until then. Please note that we will not be able to formally accept your manuscript and schedule it for publication until you have completed any requested changes.

PRESS

Sincerely,

Luke

Lucas Smith, PhD

Senior Editor

PLOS Biology

lsmith@plos.org

--on behalf of--

Christian Schnell, PhD,

Senior Editor

PLOS Biology

cschnell@plos.org